

# Full-azimuthal imaging-DOAS observations of $NO_2$ and $O_4$ during CINDI-2

Enno Peters[1,2], Mareike Ostendorf[1], Tim Bösch[1], André Seyler[1], Anja Schönhardt[1], Stefan F. Schreier[3], Jeroen Sebastiaan Henzing[4], Folkard Wittrock[1], Andreas Richter[1], Mihalis Vrekoussis[5,6,7], and John P. Burrows[1]

[1]Institute of Environmental Physics (IUP), University of Bremen, Germany
[2]Institute for the Protection of Maritime Infrastructures, German Aerospace Center (DLR), Bremerhaven, Germany
[3]Institute of Meteorology, University of Natural Resources and Life Sciences, Vienna, Austria
[4]Netherlands Organisation for Applied Scientific Research (TNO), Utrecht, The Netherlands
[5]Laboratory for Modeling and Observation of the Earth System (LAMOS), Institute of Environmental Physics (IUP), University of Bremen, Germany
[6]Center for Marine Environmental Sciences (MARUM), University of Bremen, Germany
[7]Energy, Environment and Water Research Centre, The Cyprus Institute (CyI), Nicosia, Cyprus

*Correspondence to:* Enno Peters (Enno.Peters@dlr.de)

**Abstract.** A novel imaging-DOAS instrument (IMPACT) is presented combining full-azimuthal pointing ($360°$) with a large vertical coverage ($\sim40°$). Complete panoramic scans are acquired at a temporal resolution of $\sim15$ minutes enabling the retrieval of $NO_2$ vertical profiles over the entire panorama around the measurement site.

IMPACT showed excellent agreement (correlation >99%) with coinciding MAX-DOAS measurements during the CINDI-2
campaign. The temporal variability of $NO_2$ slant columns within a typical MAX-DOAS vertical scanning sequence could be resolved and was as large as 20% in a case study under good viewing conditions. The variation of corresponding profiles and surface concentrations were even larger (40%). This variability is missed when retrieving trace gas profiles based on standard MAX-DOAS measurements.

The azimuthal distribution of $NO_2$ around the measurement site showed inhomogeneities (relative differences) up to 120%
(on average 35%) on short time scales (individual panoramic scans). This is more than expected taking into account the semi-rural location. One reason for this are transport events. Exploiting the instrument's advantages, the plume's trajectory during a prominent transport event could be reconstructed.

Furthermore, the potential of retrieving information about the aerosol phase function from $O_4$ slant columns along multiple almucantar scans of IMPACT is demonstrated, with promising results for future studies.

# 1 Introduction

Nitrogen dioxide ($NO_2$) is a prominent pollutant in the atmosphere and harmful for human health causing damage to the respiratory system (Kampa and Castanas, 2008). It originates from NO that is produced in the equilibrium between $N_2$ and





$O_2$ at high temperatures in combustion processes. The emitted NO reacts with ozone ($O_3$) to form $NO_2$. The sum of NO and $NO_2$ is called $NO_x$.

The UV photolysis of $NO_2$ produces NO and O atoms, which react with $O_2$ in air to form $O_3$. Under certain conditions for $NO_x$ and $O_3$ in the troposphere, the Leighton photo-stationary state is achieved:

$$\frac{[NO]}{[NO_2]} = \frac{J(NO_2)}{k(NO + O_3)[O_3]} \tag{1}$$

where $J(NO_2)$ is the photolysis frequency for $NO_2$ in an air mass and $k(NO+O_3)$ is the rate coefficient for the reaction of NO with $O_3$. Deviation from the Leighton photo-stationary state occurs when $NO_2$ is produced by reaction of hydroperoxyl radicals ($HO_2$), or organic peroxy radicals ($RO_2$), with NO (e.g., Shetter et al., 1983). The photolysis of this $NO_2$ then results in the $O_3$ formation, as found in photochemical smog. Thus, $NO_x$ plays a key role in the formation of tropospheric ozone.

Emission sources of $NO_x$ are both, anthropogenic, predominantly due to the combustion of fossil fuels for domestic heating and cooking, in industry, for power generation and by traffic fleet, as well as biogenic, e.g. from savanna and forest fires. $NO_x$ is also released from lightning events and soil microbial processes (Lee et al., 1997).

Overall the lifetime of $NO_2$ in the atmosphere is typically in the order of several hours due to photolysis or removal by OH, which leads to the formation of $HNO_3$ and thus contributes to acidification of precipitation, soil and water. $NO_2$ shows characteristic absorption bands in the UV and visible wavelength range facilitating quantification by differential optical absorption spectroscopy (DOAS) measurements.

DOAS is a well-established remote sensing technique used for atmospheric trace gas observations, which arguably reaches back to Dobson and Harrison (1926) who detected stratospheric ozone using UV measurements at distinct wavelengths. Later, Brewer et al. (1973) and Noxon (1975) used zenith-sky pointing measurements of scattered sunlight to retrieve stratospheric $NO_2$ abundances. Perner et al. (1976) and Platt et al. (1979), who first used the term *DOAS*, applied active DOAS for measurements of further trace gases in the troposphere using artificial light sources. The passive DOAS technique was continuously improved to so-called *off-axis* (1D) and 2D-pointing instruments (Hönninger et al., 2004, provide a brief historic overview about passive DOAS systems) and recently even 3D MAX-DOAS analysis techniques have been reported (Ortega et al., 2015; Seyler et al., 2018). In addition to static platforms, passive DOAS was also adopted to movable platforms, e.g. cars, ships, airplanes (e.g., Sinreich et al., 2010; Shaiganfar et al., 2011; Peters et al., 2012) as well as satellites (e.g., Burrows et al., 1999; Richter et al., 2005; Lelieveld et al., 2015).

In this study, the DOAS method has been combined with imaging capabilities. Pushbroom imaging-DOAS instruments consisting of a spectrometer equipped with a 2D CCD or CMOS camera are often used for aircraft applications (Heue et al., 2008; Popp et al., 2012; Schönhardt et al., 2015). The spectrometer's slit and thus the spatial axis of the spectrometer/CCD-system is aligned perpendicular to the flight direction while pixel size along track is determined by the integration time and aircraft speed. Imaging DOAS instruments have been also used in ground-based applications. Lohberger et al. (2004) observed the $NO_2$ plume emitted from a power plant stack by using an imaging spectrometer mapping different elevation angles on the vertical (spatial) axis of the CCD and a motorized mirror system for scanning in the azimuthal direction. The same instrumental setup was used by Bobrowski et al. (2006) to observe the $SO_2$ emission from a volcano. A scanning mirror system was also



used by Lee et al. (2009) to analyze the spatial and temporal variation of $NO_2$ during two days in the urban environment of Beijing.

Another imaging-DOAS concept was recently described by Manago et al. (2018) consisting of a combination of horizontal slit, transmission grating and hyperspectral camera acting effectively as a line scanner to produce a $13°x9°$ image with spectral

information. 87 hyperspectral images were combined during an acquisition time of $\approx$ 1 hour to a full-azimuthal panoramic view in order to study the two-dimensional $NO_2$ distribution around the measurement site.

In summary, all reported imaging-DOAS observations have in common that a very small angular resolution was applied resulting in a rather limited total field of view (FOV) for the entire image (e.g., $13°x36°$). While this approach is valuable for example for the observation of the trace gas emitted from a power plant or volcano, the observed scene is limited in its

spatial scale. In contrast, the aim of the instrument concept presented in our study is to provide full azimuthal coverage (360°) around the measurement site with, at the same time, a large vertical coverage ($\sim$40°). Aiming at high robustness and flexibility (predominantly for separating outdoor and indoor parts), no scanning mirror system but a telescope with a sorted quartz fibre bundle pointing in several elevations at the same time, and a pan-tilt-head for scanning in the azimuthal direction, are used. This setup enables profile retrievals in the entire hemisphere around the instrument at sufficiently high temporal resolution and to

study the full 2-dimensional distribution and variability. The short acquisition time of a full panoramic image ensures constant atmospheric conditions and thus, minimizes the impact of temporal changes of trace gas distributions during the observation.

The imaging DOAS instrument IMPACT (novel Imaging MaPper for AtmospheriC observaTions) took part in the CINDI-2 campaign in Summer 2016, where it participated in the semi-blind intercomparison of $NO_2$. Results of the intercomparison are not a primary focus of this study and are presented in detail in (Kreher et al., 2018).

The main objective of the present study is to assess the added value of full-panoramic imaging-DOAS measurements as compared to MAX-DOAS. In particular, the change of $NO_2$ profiles and surface concentrations during a typical MAX-DOAS vertical scanning sequence could be resolved. Furthermore, assessment of the azimuthal distribution of $NO_2$ is a prerequisite for satellite validation, as a point measurement (in situ) or measurements in one azimuth direction only is not representative for the entire measurement's surrounding (satellite pixel) if the azimuthal distribution is inhomogeneous. In the current study,

large inhomogeneities occurred on short timescales and were caused by transport events rather than persistent inhomogeneities (e.g. due to local sources). Due to the full-panoramic coverage, an exemplary transport event could be observed by the temporal evolution of $NO_2$ profiles. The plume's trajectory could be reconstructed and its most likely emission source was identified. In addition, information with respect to the aerosol phase function was derived from the retrieved azimuthal distribution of the $O_2$ collision complex $O_4$, which was retrieved during the DOAS fitting process in the selected spectral window used for $NO_2$.

We note that IMPACT measures simultaneously multiple almucantars[1].

The paper is structured as follows: Section 2 briefly describes the performed DOAS measurements, instruments, and the CINDI-2 campaign. The pointing calibration and FOV definition of IMPACT are explained in Sect. 3, which also contains

---

[1]Note, an *almucantar* is a circle on the celestial sphere parallel to the horizon. The almucantar containing the sun, i.e. having the sun's elevation, is the *solar almucantar*. Within the community, both terms are frequently used synonymously, but it is important to distinguish here because IMPACT measures in many elevations at the same time, i.e. records many almucantars when measuring in different azimuths.





a comparison with MAX-DOAS data. The spatial and temporal $NO_2$ variation observed during CINDI-2 is discussed in Sect. 4 including a detailed analysis of an observed transport event. Section 4 also discusses the potential of retrieving aerosol phase function information from IMPACT's observations. A subsequent profile retrieval based on the full-panoramic measurement strategy was used to further investigate the $NO_2$ distribution around the measurement site, in particular during the transport

event. The study closes with a summary and conclusion.

## 2   Measurements

### 2.1   DOAS technique

The passive DOAS technique uses measurements of scattered sunlight and the Lambert-Beer's law to yield trace gas amounts and distributions in the atmosphere. While scattering causes smooth changes in the spectrum (e.g., $\lambda^{-4}$-dependence for

Rayleigh scattering), molecular absorption often has structured spectra. The total spectral attenuation is therefore split into a high-frequency part comprising the trace gas absorptions and a low-frequency part accounting for elastic scattering on molecules, aerosols, and clouds, as well as instrumental throughput. This part is described by a low-order polynomial. The effect of inelastic scattering, which is predominantly due to Rotational-Raman-Scattering known as the Ring effect (Shefov, 1959; Grainger and Ring, 1962) is accounted for by a pseudo cross section (e.g. Vountas et al., 1998). Lambert-Beer's law can

then be expressed by the DOAS equation:

$$\tau = \ln\left(\frac{I_0}{I}\right) = \sum_i \sigma_i \cdot SC_i + \sigma_{\mathrm{Ring}} \cdot SC_{\mathrm{Ring}} + \sum_p a_p \lambda^p + r \qquad (2)$$

where $\tau$ is the optical depth and the first sum is over all absorbers $i$ having cross sections $\sigma_i$. The polynomial degree is $p$, and the residual term $r$ contains the remaining (uncompensated) optical depth, for example from measurement noise.

As measurements consist of spectra $I$ and $I_0$, Eq. 2 is defined at many wavelengths and solved in a linear least-squares fit

returning the fit factors $SC_i$ and $a_p$. While the polynomial coefficients $a_p$ are usually not used for further analysis, the so-called slant columns $SC_i = \int \rho_i ds$ are the integrated concentration $\rho_i$ of absorber $i$ along the light path $s$.

Recorded spectra contain almost no information about the altitude, in which the absorption occurred. Thus, the sensitivity to different altitudes depends predominantly on measurement geometry. The measurement is more sensitive to tropospheric absorbers, if the spectrum $I$ is taken at small elevation angles above the horizon. This is due to the rather long light path

through atmospheric layers close to the surface. On the other hand, the reference spectrum $I_0$ is usually a zenith spectrum either at small sun zenith angle (SZA), or taken close in time to the measured spectrum $I$ (sequential), as the light path through the atmosphere is short then. The obtained $SC_i$ are therefore not absolute but the difference between measurement ($I$) and reference measurement ($I_0$) and thus called differential slant column density (DSCD). As only DSCDs are used within this study, both terms are used synonymously in the following for simplicity. Furthermore, sequential reference fits are used

throughout this study.

More details of the DOAS method can be found for example in (Hönninger et al., 2004; Platt and Stutz, 2008).



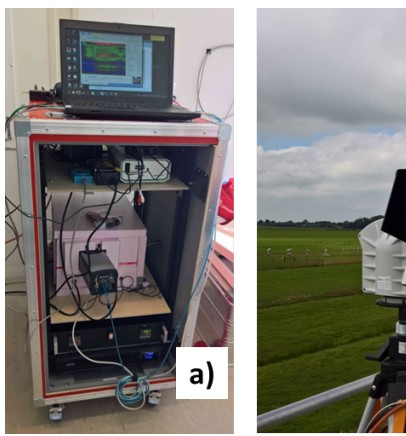
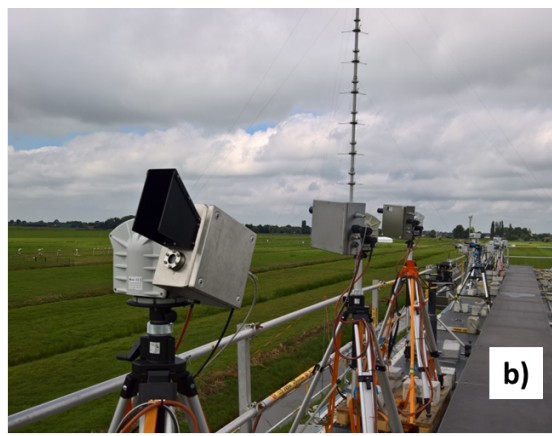

**Figure 1.** The IMPACT instrument installed during CINDI-2. a) Indoor parts integrated into a 19" rack. b) Telescope unit on top of the container deck (foreground). Next to IMPACT is the IUP-Bremen 2D-MAX-DOAS instrument (background) used for comparison in Sect. 3.2.

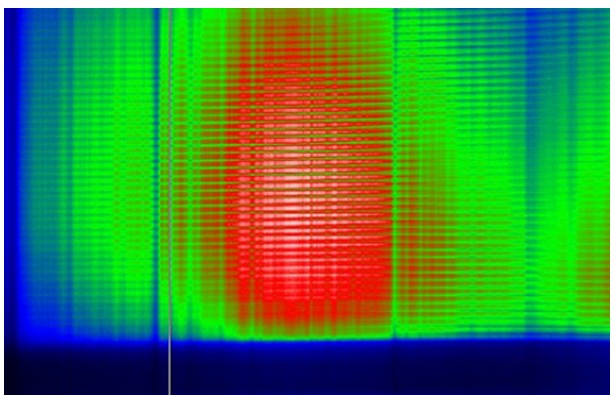

**Figure 2.** Typical CCD image as recorded during CINDI-2. The x-axis is the spectral direction while the y-axis represents the viewing elevation. Blue areas indicate low illumination while green, red and white colours represent increasing light levels. The x-axis covers 394.5-536.4 nm, i.e. for the DOAS fit of 425-490 nm only the inner part is used. On the y-axis, single fibres observing different elevation angles are separated and distinguishable. Fraunhofer lines are visible in each fibre at the same spectral position. The horizon causes a sharp transition between illuminated and non-illuminated fibres in the lower part of the image.

## 2.2  IMPACT

The IMPACT instrument consists of a Czerny-Turner type ANDOR Shamrock 303i imaging spectrometer equipped with a Newton DU940P-BU CCD camera with 2048x512 pixels covering a wavelength range from 394.5-536.4 nm. The CCD is cooled to -30°C for reducing the dark signal (thermal electrons), while the spectrometer is actively temperature stabilized to +35°C in order to avoid thermal (and therefore spectral) drifts. The spectrometer-CCD-system is installed within a 19" rack that hosts at the same time all electronics and computers for instrumental control and operation. A 15 m long light fibre bundle

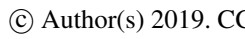



consisting of 69 individual fibres (0.01 mm$^2$ each) separates the indoor part (rack) from the telescope unit located outside. At both sides, the individual fibres are aligned vertically, i.e. stacked on top of each other (total height ∼9 mm), and sorted in a way that the uppermost fibre on the entrance side is also the uppermost fibre on the spectrometer side. However, light from the upper- and lowermost fibres do not hit the detector (these fibres are imaged outside the detector area), so that only 50 individual

fibres fully mapped on the CCD are used here.

Light is collected and focused on the light fibre bundle with a commercial objective (1:1.4, focal length 8 mm) resulting in a hypothetical vertical FOV of ∼58° for the entire fibre bundle, i.e. all 69 stacked single fibres. The finally utilized part of the measurements yields a vertical FOV of ∼41°. The use of an objective instead of a single lens is necessary for overcoming spherical aberration and thus keeping the FOV constant for each individual fibre as the entrance slit has a considerable height

(9 mm). This is different to usual MAX-DOAS instruments where the light is focused on a very small spot-sized fibre entrance located on the optical axis and therefore using a single lens is usually sufficient.

The vertical alignment of the sorted light fibres in combination with an imaging spectrometer - each fibre is mapped onto different CCD lines - allows to take measurements in multiple elevation angles simultaneously (see Sect. 3.1 for the calibration procedure of the elevation angle). Furthermore, the telescope hosts a visual camera taking snapshots for scene documentation

with each measurement. The telescope unit is installed on an ENEO VPT-501 pan-tilt head, which allows pointing in any direction. However, as a result of the sufficiently large instantaneous vertical FOV, movements are performed in azimuthal direction only while the vertical tilt is kept constant (covering the elevation angles from -5° to +36°) with the exception of zenith-pointing for taking reference measurements. The IMPACT instrument as deployed during the CINDI-2 field campaign (Sect. 2.4) is depicted in Fig. 1.

Figure 2 shows an example image of the CCD for a typical off-axis measurement. The image quality (separation of single fibres) is best in the center of the CCD and blurred towards the edges. This is because the horizontal (spectrometric axis) and vertical (spatial axis) foci do not coincide everywhere in the focal plane (coincidence is optimized for the center of the CCD). The CCD can be placed in different positions, resulting either in good imaging or good spectrometric quality. Here, an intermediate flange was used placing the CCD in a position that is a compromise between imaging and spectroscopic

performance. As a result, the slit function changes vertically across the detector from ≈1 nm FWHM in the center of the CCD to ≈1.5 nm FWHM towards bottom and top rows. This was compensated for in the DOAS analysis by measuring and applying separate slit functions for different vertical binning ranges on the CCD associated to individual light fibres as defined in Sect. 3.1.

Ideally, an imaging instrument should be operated with a shutter or a frame transfer CCD in order to minimize the impact of

illumination of the detector during readout. As the Newton DU940P-BU is not a frame transfer CCD and long-term operation of a shutter is limited by shutter lifetime, IMPACT measurements are taken without a shutter. As a result, the detector continuous to be illuminated during the sequential CCD-readout, leading to larger signals in those rows which are read out later. As the vertical position on the CCD corresponds to different elevation angles, this leads to a smearing of the CCD image and the corresponding viewing directions.



**Table 1.** DOAS fit settings for $NO_2$ and $O_4$.

| Parameter | Value |
| --- | --- |
| Reference ($I_0$) | sequential |
| Fit window | 425-490 nm |
| Polynomial | degree 5 |
| Intensity offset correction | Offset (zeroth order) |

| Cross-section | Reference |
| --- | --- |
| $O_3$ | (Serdyuchenko et al., 2014) at 223 K with $I_0$-correction (SC of $10^{20}$ molec/cm$^2$) |
| $NO_2$ | (Vandaele et al., 1996) at 298 K and 220 K (orthogonalised to 298 K) |
| | with $I_0$-correction (SC of $10^{17}$ molec/cm$^2$) |
| $O_4$ | (Thalman and Volkamer, 2013) |
| $H_2O$ | HITEMP (Rothman et al., 2010) |
| Ring | QDOAS (provided during CINDI-2) |

If illumination is assumed to be constant during measurements, a simple correction can be applied to the measured data. Starting from the very first line for which there is no smear effect, the original signal can be computed for each line successively by subtracting the additional illumination occurring during readout:

$$I_j = I_j^{meas} - \sum_{k=1}^{j-1} I_k \cdot \frac{t_{\text{readout}}}{t_{\text{exposure}}} \tag{3}$$

where $I_j$ is the signal of row $j$ without smear, $I_j^{meas}$ is the intensity with smear, and $t_{\text{readout}}$ and $t_{\text{exposure}}$ are the length of the duration of the readout of one line and the exposure time, respectively. While this correction works well in most cases, it can fail in situations where illumination changes rapidly, for example during measurements with broken clouds and high wind speeds.

Problems regarding the smear effect generally decrease with the ratio of exposure time to readout time because the relative
contribution of illumination during readout then decreases. In other words, $I_j$ approaches $I_j^{meas}$ for $t_{\text{readout}}/t_{\text{exposure}} \to 0$, see Eq. 3. To take advantage of this, an optical filter blocking parts of the sunlight was installed in the telescope unit. This allowed to increase exposure times while avoiding saturation of the CCD.

## 2.3   MAX-DOAS instrument (IMPACT validation)

Data of the IUP-Bremen MAX-DOAS instrument is used to validate corresponding IMPACT measurements (see Sect. 3.2).
Both instruments were set-up side by side (~2 m distance, see Fig. 1). The MAX-DOAS instrument consists of a telescope unit (outdoor) and two CCD-spectrometer systems measuring in the UV and visible (indoor), respectively. For validation of IMPACT observations (measuring in the visible), only data collected by the visible spectrometer is used, which is an ACTON-



500 covering a spectral range from 406-579 nm at a resolution of ≈0.85 nm. The spectrometer was actively temperature stabilized to +35°C. A Princeton NTE/CCD 1340/100-EMB with 1340x100 pixels was used for recording spectra leading to a spectral sampling of 7-8 pixels/nm. The CCD was cooled to -30°C to reduce dark signal.

Light was collected by a telescope unit mounted (similar to IMPACT) on a commercial ENEO VPT-501 pan-tilt head
allowing pointing in any viewing direction. The instrument's FOV ($\approx 1.1°$) was determined by a lens focusing incoming light on an optical fiber bundle (length ≈20 m), which was Y-shaped and connected the telescope with both spectrometers. It consists of 2x38 = 76 single fibres. An in-telescope shutter and HgCd line lamp allow dark and wavelength-calibration measurements, which were routinely performed. A very similar instrumental set up has been used in previous campaigns, e.g. CINDI and TransBrom (Roscoe et al., 2010; Peters et al., 2012).

**2.4   The CINDI-2 field campaign**

The Second Cabauw Intercomparison of Nitrogen Dioxide measuring Instruments (CINDI-2) field campaign was carried out at the Cabauw Experimental Site for Atmospheric Research (CESAR), close to the villages of Cabauw and Lopik, the Nether-lands, from 25 August to 7 October 2016. It was a successor of the first CINDI campaign in 2009 (Roscoe et al., 2010; Piters et al., 2012). CINDI-2 aimed at characterizing the differences between measurement approaches and systems and to progress
towards harmonization of settings and methods (Hendrick et al., 2016). One key activity was a semi blind intercomparison of participating DOAS-type instruments from different international research groups. This intensive phase was scheduled for the time period 12-25 September 2016.

The measurement test site is located in a semi-rural environment, i.e. without strong local sources (except for a regional traffic road in the South potentially causing enhanced $NO_x$ levels during rush hour) but within the polluted region between
Amsterdam, Rotterdam and Utrecht.

In total, 23 groups and 31 DOAS-type instruments participated in CINDI-2. The instruments were mainly deployed at two container decks. At the lower level, 1D MAX-DOAS instruments were pointing permanently in a common azimuth direction of 287° (clock-wise from North) and performed vertical scanning sequences in this azimuth. 2D MAX-DOAS systems in-stalled at the upper container deck (see Fig. 1) providing a free view around the measurement site were following a rather
complex measurement protocol (Kreher et al., 2018) prescribing the observation geometry on a 1-minute timebase. However, for comparison with 1D instruments, a vertical scanning sequence was performed in the common azimuthal direction every hour.

The IMPACT instrument fulfilled two purposes during CINDI-2:

1. To participate in the semi-blind intercomparison. For this reason, measurements were performed in the common azimuth
30       direction of 287° every hour, together with the 1D- and 2D-instruments.

2. To study the added value of full-panoramic imaging measurements at high repetition rate, in particular for estimating the spatial distribution and its temporal variability around the measurement site. Therefore, between hourly intercomparison measurements, full-azimuthal scans in 10° steps were taken. For each azimuth direction, a complete set of elevation





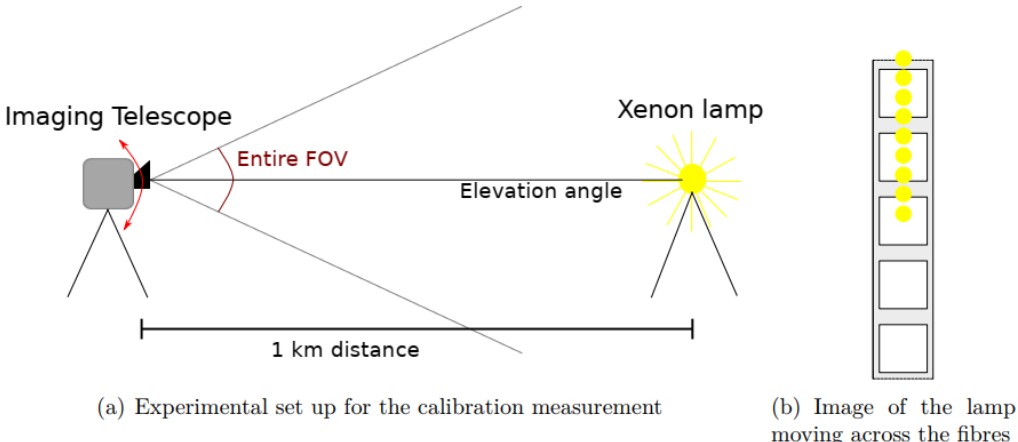

**Figure 3.** Scheme of calibration measurement procedure (Ostendorf, 2017).

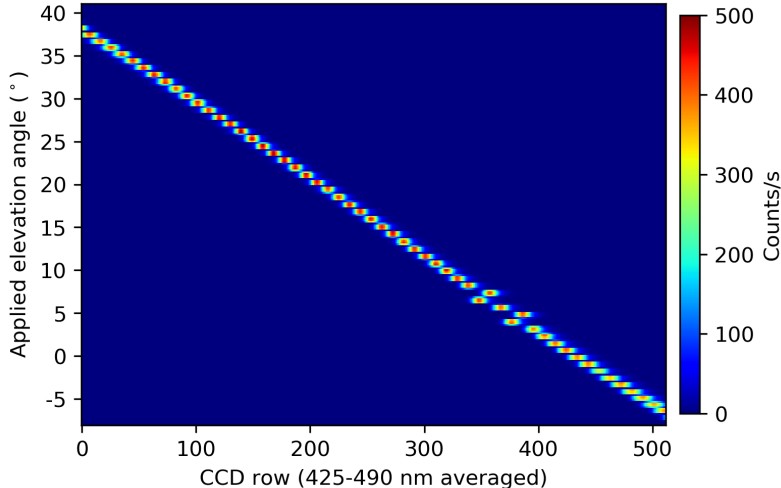

**Figure 4.** Elevation angle calibration matrix: The intensity in the fitting range is displayed as function of the CCD row (x-axis) and telescope elevation angle (y-axis).

angles was observed simultaneously due to the imaging capability of the system. As a result, a full panoramic view (-175° to 175° azimuth in 10° steps and ≈-5° to 36° elevation in ≈ 0.8° steps) was recorded every 15 minutes.

In addition to the observation geometry, also DOAS fit settings were prescribed for the CINDI-2 semi-blind intercomparison (Tab. 1). These fit parameters have been used as well for the analysis of $NO_2$ and $O_4$ distributions within this study.





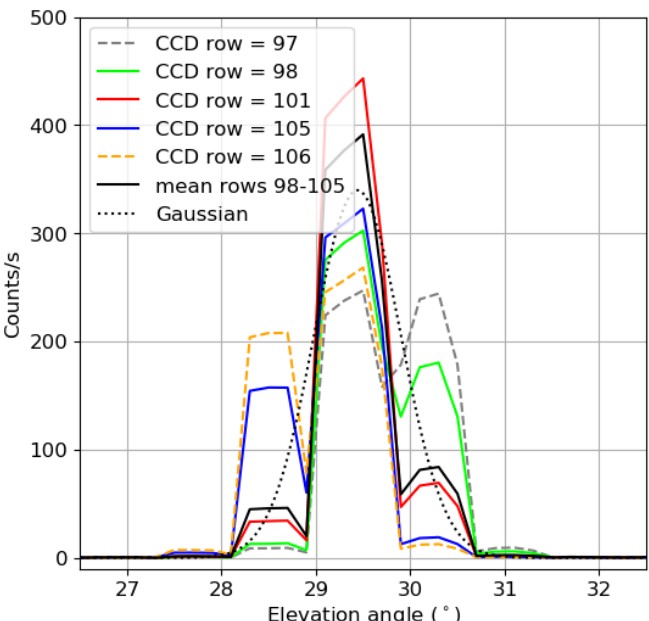

**Figure 5.** Cross sections through the calibration matrix. The defined binning range comprises rows 98-105 which all show a clear maximum in the same fibre, most pronounced in row 101. CCD rows 97 and 106 are rejected as their intensity distribution cannot be clearly assigned to one fibre. The mean of the binning range is plotted in black together with the corresponding Gaussian curve (same standard deviation) in order to estimate the effective FOV.

## 3   Instrument performance

### 3.1   Pointing calibration

The calibration of the elevation angles in which IMPACT is taking measurements simultaneously was performed on-site during CINDI-2 as part of a pointing calibration exercise that was organized by the Max-Planck Institute for Chemistry (MPIC),

5   Mainz, who operated a Xenon lamp positioned in a distance of ≈1 km from the measurement site (many thanks in particular to Sebastian Donner, Jonas Kuhn, and Thomas Wagner). Details about the exercise can be found in (Donner et al., 2019).

Fig. 3 shows a sketch of the experimental setup. IMPACT's telescope was moved in 0.2°-steps across the Xenon lamp. It is important to realize that changing the elevation angle moves the image of the lamp across the fibre entrances in the telescope while the imaging of individual fibres on the CCD is independent of telescope elevation. For each measurement,

10   only one individual fibre was illuminated meaning that the spot of the Xenon lamp at the light fibre entrance was smaller than the diameter of a single fibre (Fig. 3b). Furthermore, each fibre was illuminated for ≈4 steps before the signal was promptly switching into the neighbouring fibre in the following measurement. This indicates an instantaneous FOV of ≈0.8° for single fibres (which is determined by the objective and the size of the fibres).



In Fig. 4, the intensity of each CCD row (averaged in the spectral fitting region between 425-490 nm) is shown as a function of telescope elevation angle. As can be seen from this calibration matrix, the (vertical) extent of a single fibre mapped onto the CCD is typically $\approx$19 CCD rows (x-axis in Fig. 4) with the tendency of smaller extents in the center and larger extents towards the edges. This is caused by better imaging quality in the center of the CCD as mentioned before. However, the spacing

between intensity maxima is only $\approx$9 CCD rows meaning that images of different individual fibres overlap each other (due to the limited imaging quality of the spectrometer). The overlapping is larger towards the edges and smaller in the center.

The pointing calibration procedure consists of 3 steps:

1. CCD rows corresponding to the same fibre were identified and binned. For this, each vertical cross section of the calibration matrix (i.e. each CCD row) was analyzed as shown in Fig. 5. CCD rows having a distinct maximum for the

same fibre were binned while CCD rows having no clear maximum were rejected (as a criterion for a distinct maximum, a ratio of at least 1.5 between the intensity in different fibres was used). However, the assignment between CCD row and elevation angle is still not unique due to the overlapping of fibre images on the CCD. This results in an effective FOV which is larger than $0.8°$ (see below).

2. An intensity-weighted elevation angle is calculated for each CCD row:

$$\text{Weighted elevation}_i = \frac{\sum_i \text{intensity}_i \cdot \text{elevation}_i}{\sum_i \text{intensity}_i} \tag{4}$$

where $i$ is varied over all applied elevation angles.

3. The weighted elevations are then averaged according to the binning intervals.

In this way, 50 binning ranges and corresponding elevation angles were defined in which measurements are performed simultaneously.

The effective FOV (per binning range) was estimated by the FWHM (full width at half maximum) of Gaussians having the same standard deviation as the weighted elevation angles (calculated in step 2) within the respective binning range. For the example shown in Fig. 5, an effective elevation of $29.4°$ and a FOV of $1.1°$ is obtained.

A prominant feature in Fig. 4 are two pairs of permuted individual fibres. This was discovered on-site only and is a defect of the fibre bundle used which was corrected by the manufacturer after the campaign. However, as a result of the performed

calibration procedure, the effective elevation assigned to the twisted fibres is correct. The effective FOV is approximately twice as large as for the other viewing directions because fibres which are next to each other at the spectrometer entrance and contribute due to the overlap are not properly ordered on the telescope side and therefore not pointing in adjacent elevation angles.

## 3.2   Intercomparison to MAX-DOAS measurements

Figure. 6 shows $NO_2$ DSCDs from an example MAX-DOAS vertical scanning sequence on 23 September 2016 under good weather and viewing conditions in comparison to IMPACT results. Note that due to instrumental restrictions, the elevation



**Table 2.** Statistics (correlation coefficient, slope and offset) between IMPACT and MAX-DOAS $NO_2$ slant columns from Fig. 7.

| Elevation | Correlation | Slope | Offset ($10^{15}$ molec/cm$^2$) |
|---|---|---|---|
| 2° | 0.995 | 0.99 | 4.76 |
| 5° | 0.998 | 1.03 | 0.59 |
| 15° | 0.997 | 1.08 | -0.53 |
| 30° | 0.979 | 1.07 | 0.42 |

angles of IMPACT deviate slightly from the angles prescribed for the semi-blind intercomparison, while the MAX-DOAS instrument follows exactly the prescribed angles. As a result, the column for the 1° MAX-DOAS elevation (blue triangle) should be slightly larger than the IMPACT slant column (blue circles) taken at the same time because the effective elevation of IMPACT is 1.4°. Interestingly, this is not seen here (the $NO_2$ slant columns of both instruments agree quite well). The reason might be small misalignments between both instruments, either in elevation or azimuth, or the $NO_2$ profile shape (potentially in combination with differences in the FOV of both instruments).

Fig. 6 demonstrates a striking advantage of imaging-DOAS as measured $NO_2$ slant columns reveal a short-term temporal variation, which is resolved by IMPACT but not by the MAX-DOAS instrument. As mentioned, the 1° MAX-DOAS observation matches the IMPACT observation taken at the same time, but then MAX-DOAS continues with the next elevation (2°) while IMPACT repeats measurements of the complete elevation angle range. In the case of 1° (1.4°) elevation, the $NO_2$ slant columns change from $\sim 1.75 \cdot 10^{17}$ molec cm$^{-2}$ to $\sim 1.40 \cdot 10^{17}$ molec cm$^{-2}$, which is about 20%. This temporal variation is not captured by the MAX-DOAS instrument, with clear consequences for any profile retrieval on these data which assumes that measurements at different elevation angles probe the same atmosphere. This is further investigated in Sect. 4.3.

Figure 7 shows the correlation plot between MAX-DOAS and IMPACT $NO_2$ slant columns for several days within the semi-blind intercomparison phase. For each MAX-DOAS elevation angle (color-coded) the closest IMPACT vertical scan (measured simultaneously) was selected. As a quality criterion, data was rejected if no IMPACT scan was found $\pm$ 2 minutes around the MAX-DOAS measurement time (e.g. due to instrumental failures or saturated data). In addition, $NO_2$ slant columns from IMPACT's simultaneous elevations were interpolated to the MAX-DOAS elevation angle. Statistical values for the correlation plot are summarized in Tab. 2. In general, an excellent agreement is found with correlation coefficients of $\approx$ 98% for 30° elevation angle and even > 99% for elevation angles $\leq$ 15°. The slope is close to 1 (within 8%) and the offset is $< 1 \cdot 10^{15}$ molec cm$^{-2}$ with the exception of the 2° elevation, for which it is slightly larger.

## 4 Results

### 4.1 Azimuthal $NO_2$ distribution and transport events

Figure 8 shows the campaign average of $NO_2$ in all azimuths and elevation angles around the measurement site. For better visibility, the 5 lowermost CCD bins (corresponding to single fibres) pointing towards the ground have been removed as

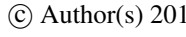



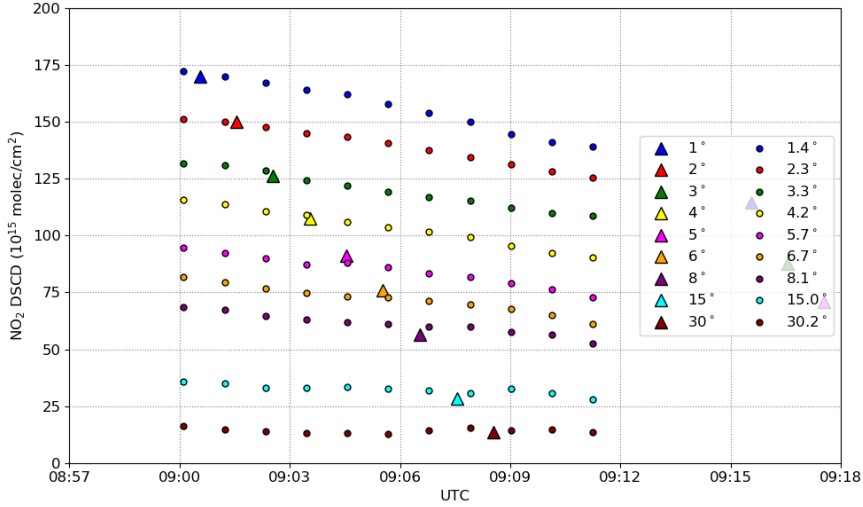

**Figure 6.** $NO_2$ DSCDs from an exemplary MAX-DOAS vertical scan (triangles) on 23 September 2016 compared to IMPACT (circles). Note, IMPACT's elevations deviate slightly from prescribed MAX-DOAS elevations.

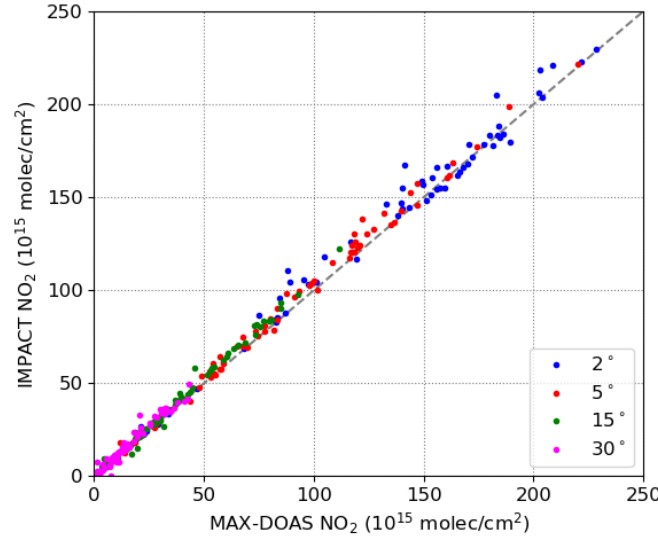

**Figure 7.** Correlation plot of $NO_2$ DSCDs from MAX-DOAS and IMPACT instrument for 17-23 September 2016 during CINDI-2. The elevation angle is color-coded, the 1:1 line is dashed.

well as 2 CCD bins pointing effectively in almost the same direction as a result of the twisted fibres discussed in Sect. 3.1. Consequently, the panoramic view in Fig. 8 consists of 43 elevation angles on the vertical axis and 36 azimuth directions (-175°





to +175° in 10° steps) on the horizontal axis. In addition, the fractional IMPACT elevation angles on the vertical axis have been rounded for better readability. We note that this has been done in subsequent figures and in the following discussion as well.

Obviously, the campaign mean $NO_2$ distribution around the measurement site is rather homogeneous with a slight tendency
to larger values in the South-West (between -165° and -75°) which is most likely linked to a close-by local traffic road (in this azimuthal regime, the light path is almost along the road, which can be seen in Fig. 12). Furthermore, the light path was obstructed by trees in ≈75° to 135° azimuth and elevation angles <5° which can be clearly seen by reduced $NO_2$ slant columns in these directions - i.e. these small values are an effect of obstacles and the resulting short light path. In addition, obstruction by other instruments occurred in -25° and by a single tree in -115°. In general, largest $NO_2$ slant columns are found not in 0°
or 1° but ≈2° elevation, which is an effect of the instrument's FOV, i.e. surface effects are present in the 0° and (to a lesser extent) in 1° elevation angle as a result of the overlap of adjacent fibres mapped onto the CCD (see Sect. 3.1 and Fig. 5).

The homogeneous long-term averaged $NO_2$ distribution around the measurement site is supporting the assumption of the absence of persistence strong local pollutants. However, much more variability is present on shorter time scales. This is demonstrated by Fig. 9a where the range of $NO_2$ slant columns recorded on 20 September 2016 (maximum and minimum values) as
well as the mean over all applied azimuths in 4° elevation angle is shown (one data point for each panoramic view). Maximum values differ from the azimuthal mean by up to a factor of 2. This is quantitatively analyzed for the whole campaign in Fig. 9b showing the maximum relative difference, i.e. the ratio between maximum $NO_2$ observed in any azimuth to the $NO_2$ averaged over all azimuths. The maximum relative differences range from 10% to 120% for individual panoramic views and are ≈35% on average. This is an unexpectedly high value indicating large spatial inhomogeneity on short time scales even for semi-rural
measurement sites like Cabauw with no large local sources and very homogeneous long-term trace gas distributions. As a result, care has to be taken if ground-based (MAX-DOAS) measurements are used for satellite validation as a single viewing direction does not necessarily provide a good estimate of the $NO_2$ columns within a satellite pixel. In this case, observations in many azimuths should be taken and averaged to reduce variability present in satellite ground pixels.

One reason for spatial inhomogeneity of $NO_2$ is the transport and passing of polluted air masses. Fig. 10 shows the temporal
evolution of $NO_2$ slant columns in all applied azimuth directions (vertical axis) and 4° elevation angle on 20 September 2016. The data gap around 14:00 UTC is due to an instrumental failure. Besides moderately enhanced $NO_2$ towards the evening, a clear transport event occurred around 10:00 UTC. Between 9:00 and 10:00 UTC, increased $NO_2$ slant columns appear in all azimuth directions between 25° and ~175° (South). Between 10:00 and 11:00 UTC, the maximum of $NO_2$ is then traveling from an azimuth angle of $\beta_1 \approx 30°$ to $\beta_2 \approx -70°$ (see geometrical considerations in Fig. 11).

The wind direction on 20 September 2016 was quite variable with low absolute wind speeds. However, the mean wind direction between 10:00 and 11:00 UTC was ≈75°. If the plume is transported by the wind, the direction of smallest distance $r$ to the measurement site is $\alpha \approx -15°$ (see Fig. 11). The assumption here is a straight trajectory $s$ (blue arrow) of the plume and thus the smallest distance $r$ (dashed line) to the measurement site is perpendicular to it. As can be seen from Fig. 10, this coincides roughly with the direction of largest $NO_2$ although slant columns have not necessarily to be largest at smallest



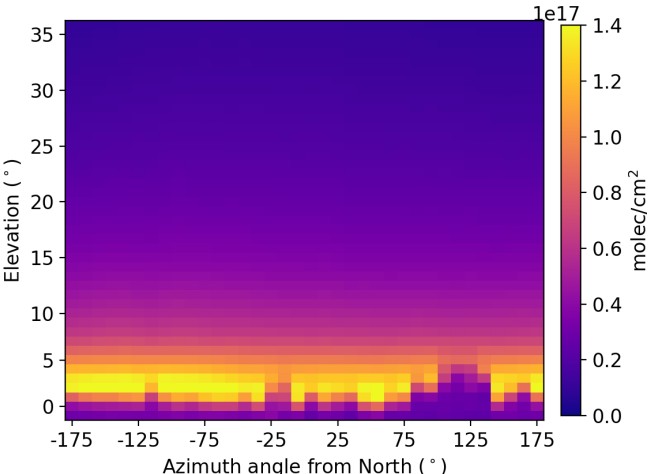

**Figure 8.** Campaign mean $NO_2$ DSCDs as a function of azimuthal and elevation angles.

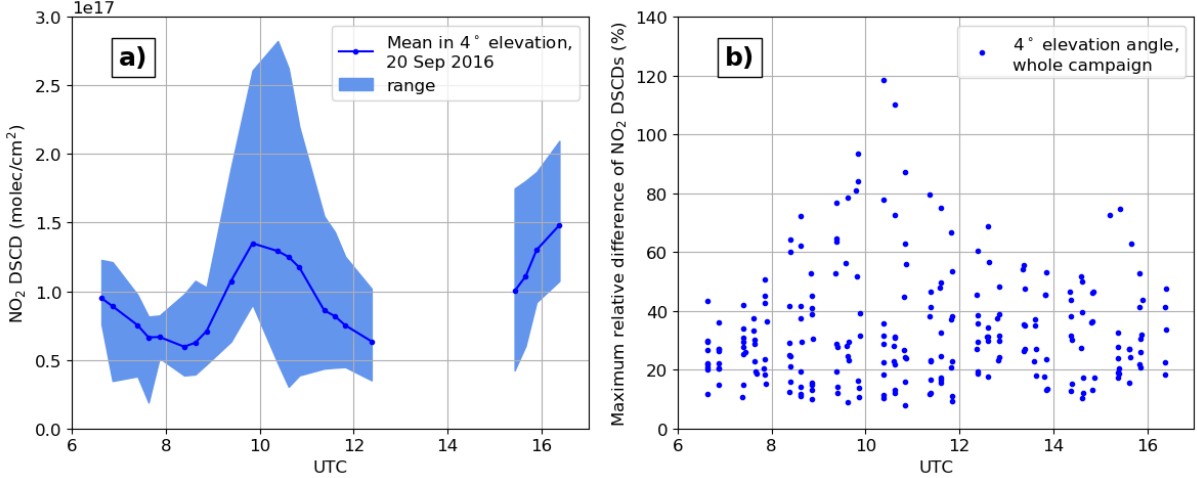

**Figure 9.** a) Range and mean of $NO_2$ DSCDs in different azimuths and $4°$ elevation angle on 20 September 2016 during CINDI-2. b) Maximum relative differences w.r.t. mean (azimuthal inhomogeneities within individual scans) for the whole campaign, as a function of UTC.

distance $r$ as the magnitude depends also for example on the (unknown) plume's shape and relative contribution of the light path through it.

The spatial distance traveled in $\Delta t = 1$ hour (10:00 to 11:00 UTC) can be estimated from wind speed:

$$s = v_{wind} \cdot \Delta t \tag{5}$$



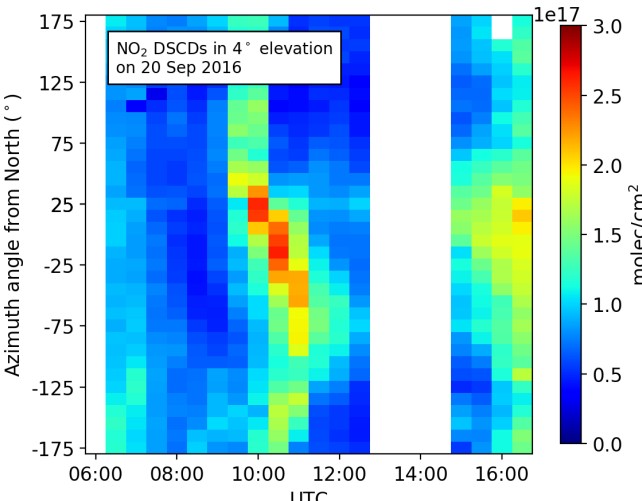

**Figure 10.** $NO_2$ DSCDs in $4°$ elevation angle (binned every 30 minutes) on 20 September 2016. A transport event occurred between 10:00 and 11:00 UTC.

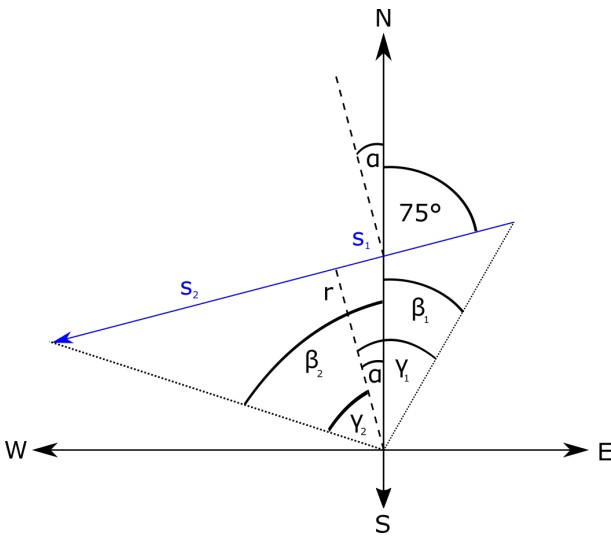

**Figure 11.** Geometry of transport event (blue arrow).

The angles between $r$ and the trajectory's start/end points (i.e. plumes's positions at 10:00 and 11:00 UTC) are $\gamma_1 = |\beta_1| + |\alpha|$ and $\gamma_2 = |\beta_2| - |\alpha|$ (Fig. 11). The distances $s_1$, $s_2$, and $s$ are then given by (omitting the sign of $\gamma_1$ and $\gamma_2$):

$$s_1 = r \cdot tan(\gamma_1) \tag{6}$$

$$s_2 = r \cdot tan(\gamma_2) \tag{7}$$

5     $$s = s_1 + s_2 = r(tan(\gamma_1) + tan(\gamma_2)) \tag{8}$$





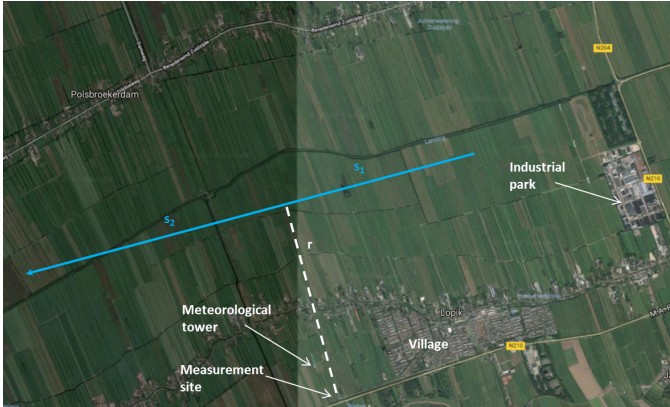

**Figure 12.** Map of the area around the measurement site. The transport event's trajectory on 20 September 2016 is indicated by a blue arrow (source: Google maps).

As a result, the smallest distance $r$ to the measurement site is:

$$r = \frac{v_{wind} \cdot \Delta t}{tan(\gamma_1) + tan(\gamma_2)} \tag{9}$$

Note that this calculation is in principle true for $0°$ elevation angle only, whereas measurements in $4°$ were used here. However, this was neglected for simplicity as the effect is small and below the uncertainty introduced by the variety of assumptions made. For a mean wind speed of 1.2 m/s measured at the Cabauw meteorological tower, a smallest distance of $r \approx 1.8$ km is obtained ($s \approx 4.3$ km, $s_1 \approx 1.8$ km, $s_2 \approx 2.5$ km).

Fig. 12 shows the measurement site's surrounding with smallest distance $r$ and plume's trajectory between 10:00 and 11:00 UTC indicated as blue arrow. Obviously, the origin of the transport event can not be precisely identified, but it could be linked to a regional industrial park that is close to the *starting point* of the plume's trajectory. This speculation is supported by the fact that increased values of $NO_2$ are already found slightly earlier ($\approx$9:30 UTC) in North-Eastern directions (see Fig. 9). In addition, increased $NO_2$ slant columns are seen in the zenith direction as well (not shown). This indicates that parts of the plume were overpassing the measurement site and thus a large spatial extent of the plume perpendicular to the direction of propagation, most likely as a result of the unstable wind direction. Finally, the fact that the $4°$ elevation angle is clearly enhanced although the plume was overpassing the instrument as well means that the plume is close to the ground which is usually an indication for a close-by origin. This is supported by vertical $NO_2$ profiles retrieved in Sect. 4.3.

## 4.2 Potential for Aerosol retrievals

In addition to $NO_2$, slant columns of the oxygen dimer $O_4$ are obtained from the same DOAS fit (Tab. 1). As $O_4$ is a collision complex of $O_2$ molecules, it depends on pressure only and is therefore a measure of the light path (e.g., Wagner et al., 2002; Wittrock et al., 2004, and references therein).





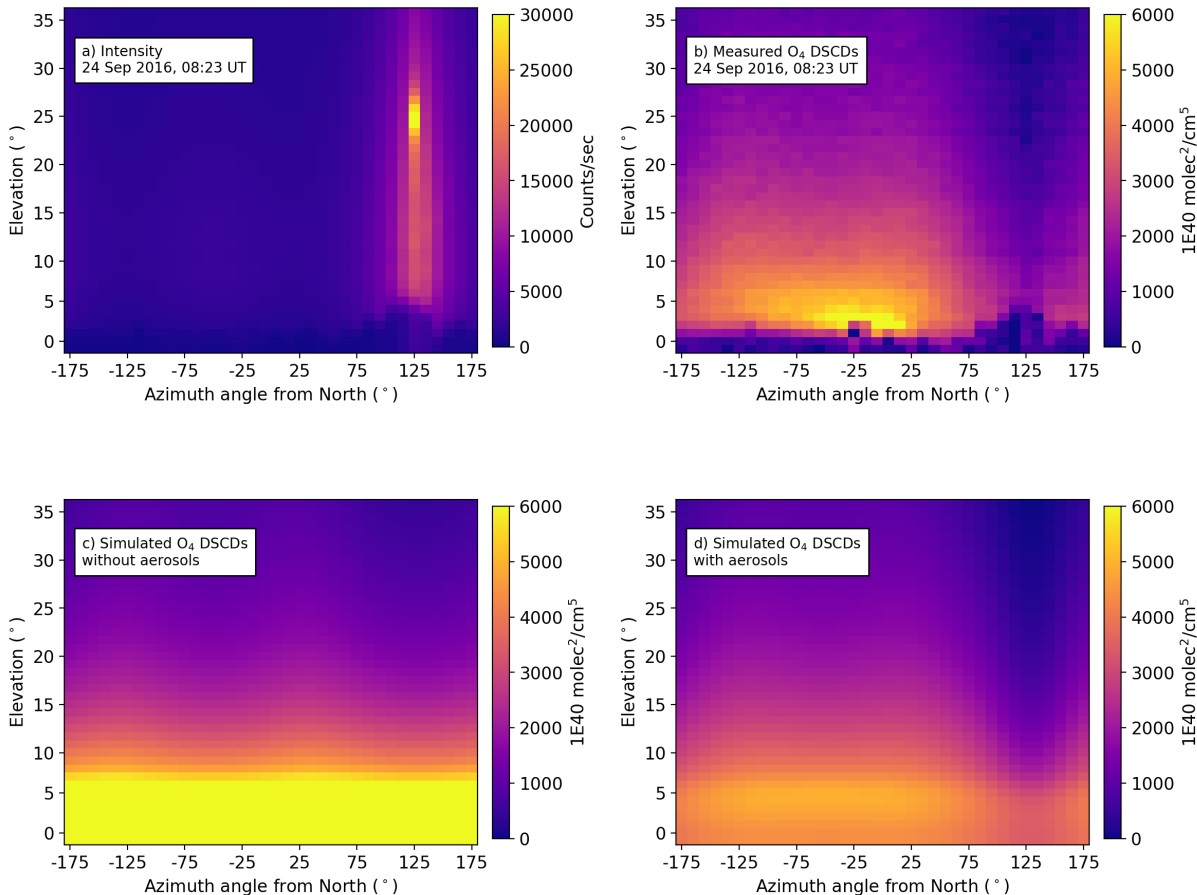

**Figure 13.** Intensity (a) and measured $O_4$ DSCDs (b) from one IMPACT panoramic scan on 24 September 2016, at 08:23 UTC mean acquisition time, in comparison to simulated $O_4$ DSCDs without (c) and with aerosols (d). Ground effects (obstacles discussed in Sect. 4.1) are of course not present in the simulations.

As a case study, Fig. 13 shows the measured intensity (a) and $O_4$ slant columns (b) from one IMPACT scan (acquisition time ~15 min) on 24 September 2016 under excellent viewing conditions. The position of the sun is clearly visible at ~125° azimuth (Solar Azimuth Angle, SAA) and ~25° elevation. $O_4$ slant columns close to the sun are reduced as a result of shorter average light paths due to strong forward scattering of aerosols. This is demonstrated by simulated $O_4$ slant columns for the same measurement geometry without aerosols, i.e. pure Rayleigh scattering (c) and with aerosols (d). The simulations have been performed using the radiative transfer model SCIATRAN (Rozanov et al., 2014) in its version 3.4.4. As input for the simulation, an exponential decrease (0.1/km surface value, $AOD = 0.2$) was used as aerosol extinction profile and a Henyey-Greenstein (HG) parameterization of the aerosol phase function with an asymmetry factor of $g = 0.75$ and a single scattering albedo $SSA = 0.95$ was applied. These values were obtained from a close-by Cabauw AERONET station (AErosol RObotic NETwork, Holben et al. (1998); Dubovik and King (2000)).

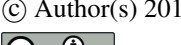



**Figure 14.** a) Measured and simulated almucantar scans of $O_4$ DSCDs in two exemplary elevation angles ($4°$ = close to the surface, and $25°$ = solar almucantar), i.e. horizontal cross sections through Figs. 13b and d. b) Same data plotted as a function of the (single) scattering angle shown in c), which has been calculated for every viewing geometry of the hemispheric scan in Fig. 13. d) Correlation coefficients between measured and simulated almucantar $O_4$ DSCDs for all elevation angles (i.e. all data from Fig. 13). Different input parameters (asymmetry factor g and single scattering albedo SSA) have been used for the simulation of $O_4$ DSCDs (for simulated data in subplots a and b, g = 0.75 and SSA = 0.95 have been used).

Simulated $O_4$ slant columns for pure Rayleigh scattering differ strongly from measured $O_4$ columns, both in absolute values as well as in the azimuthal distribution. In particular, the largely reduced columns around the sun are not reproduced by the simulation showing slightly reduced columns at the SAA and SAA + 180° only as a result of the Rayleigh phase function. In contrast, simulated columns including aerosols agree much better with measured columns and cover the azimuthal distribution



(Fig. 13d). Thus, the comparison between simulated and measured azimuthal distribution of $O_4$ columns can be used to retrieve information about the aerosol properties and in particular it's phase function.

Retrievals of aerosol properties, e.g. from AERONET stations, are usually based on intensity measurements in the solar almucantar, which in this case is the azimuthal distribution in $\approx 25°$ elevation. However, incorporating $O_4$ measurements in the retrieval of aerosol micrcophysical properties and phase function was already suggested by Wagner et al. (2004). Frieß et al. (2006) demonstrated a corresponding retrieval based on intensity and $O_4$ measurements in different azimuths and found that largest sensitivity is gained from measurements in the aureole region, therefore requiring a small FOV, protection against direct sunlight and the capability to perform automated measurements in the azimuth. While measurements very close to the sun are challenging for IMPACT due to it's large FOV, two important aspects can be investigated as a result of IMPACT's capability of recording full 2D maps very rapidly around the measurement site:

1. Is there a potential for $O_4$ measurements in almucantars different than the solar almucantar to contribute to/support aerosol retrievals?

2. Is there a restriction, which almucantars can be used, and what is the criterion/threshold for the use or rejection?

As IMPACT is (currently) not radiometrically calibrated, we focus on exploiting $O_4$ measurements rather than intensity for the retrieval of aerosol properties. In addition, it should be clearly mentioned that a full aerosol retrieval is far beyond the scope of this study, which is limited to the two research questions above.

For research question (1) it is important that sky radiometers (e.g. within the AERONET network) and current state of the art MAX-DOAS instruments are measuring in only one viewing geometry at a time. A scan along the solar almucantar then provides observations at different scattering angles. In contrast to these instruments, IMPACT measures many almucantars at the same time, in the case study shown in Fig. 13 both, above and below the solar almucantar. The geometrical scattering angle (single scattering case) has been calculated for every viewing geometry and is plotted in Fig. 14c. Obviously, almucantars above and below the solar almucantar provide slightly different scattering angles and might therefore complement the classical retrieval.

However, not all almucantars should be used and even if exploiting the solar almucantar only, a threshold for the lowest usable elevation angle should be regarded (research question 2). The reason is that a retrieval of e.g. the aerosol phase function requires the azimuthal distribution of measured $O_4$ to be caused by the aerosol phase function only. In contrast, in reality it is caused by the combined effect of 1) phase function, and 2) varying aerosol load and extinction profile in different azimuth directions as well as along the light path, i.e. in different distances from the instrument. For measurements taken at large elevations, the aerosol load and profile can be assumed to be homogeneous as the horizontal distance around the measurement site from which information is obtained (in a single scattering case this is the distance to the scattering point) is short. For small elevations, this horizontal extent around the measurement site is much larger and thus aerosol load and profile can change substantially along the light path.

This effect is clearly present in Fig. 13b: Measured $O_4$ slant columns have a distinct maximum in small elevations centered around $\approx -25°$ azimuth (ranging from $\approx -60°$ to $25°$ azimuth), which is not reproduced by simulated $O_4$ columns. As





illustrated in Fig. 14c, this is not the location of largest scattering angles occurring at $\approx -55°$ azimuth only and therefore not related to the $O_4$ maximum expected in backscattering direction (due to preferred forward scattering and consequently larger light paths in backscattering direction). Furthermore, if the $O_4$ maximum was an effect of the phase function, a second maximum would appear close to the ground at $\approx -85°$ azimuth (given that the aerosol profile would not change with the

azimuth), because scattering angles in $-25°$ and $-85°$ azimuth are identical (see Fig. 14c). Obviously, no second $O_4$ maximum is present at $-85°$ azimuth indicating that the aerosol load seen in small elevation angles changes with the viewing azimuth. In particular, the observed maximum in $O_4$ slant columns at $-25°$ azimuth indicates smaller aerosol loads close to the ground (longer light paths) in this direction. As a result, almucantar scans in small elevation angles should not be used to retrieve aerosol information.

In order to quantify this finding, Fig. 14a shows two specific azimuthal distributions of measured (solid) and simulated (dashed) $O_4$, i.e. two horizontal cross-sections of Fig. 13b and c, for elevation angles of 4°, and 25° (solar almucantar), respectively. While the agreement between measurement and simulation is very good in 25° elevation, differences in 4° are much larger, both in absolute values and in shape. Fig. 14b shows the same data, but plotted as a function of scattering angle. The solid line represents scattering angles counter-clockwise (left) from the position of the sun (SAA = 125°), and the dashed

line clockwise (right). For the solar almucantar, both lines agree quite well with each other as well as with the simulation (green line) indicating that the aerosol seen in 25° elevation is rather homogeneous around the measurement site and aerosol parameters used in the simulation are realistic. In contrast, the 4° almucantar does not match the simulation and - more importantly - $O_4$ columns observed clockwise from the incoming direction show severe differences and another shape than those recorded counter-clockwise. This cannot be explained with the aerosol phase function, which is symmetrical. It therefore

proves the conclusion of inhomogeneous aerosol content around the measurement site seen in 4° elevation.

To elaborate a threshold of usable almucantars and to test their potential for aerosol retrievals, various SCIATRAN simulations have been performed based on different aerosol parameters. For each set of parameters, resulting correlation coefficients between measured and simulated $O_4$ azimuthal distributions are shown in Fig. 14d as a function of elevation angle. Aerosol parameters leading to largest correlations are then compared to independently measured quantities from the AERONET station.

The blue curve in Fig. 14d corresponds to the original simulation shown in the previous plots using g = 0.75 and SSA = 0.95. For small elevations, correlation coefficients increase rapidly. This is due to a combination of the observed obstruction by trees discussed above and true inhomogeneities of the $O_4$ azimuthal variation. The steep increase is followed by a much shallower increase until a plateau is reached at $\approx 10°$. For very large elevations $> 30°$, correlation coefficients decrease again slightly, most likely as an effect of smaller $O_4$ columns and thus poorer statistics.

Furthermore, it is found that changes of the SSA (red line) lead to almost the same results, i.e. the pure analysis of the shape of $O_4$ columns at a specific almucantar is (not surprisingly) insensitive to the SSA.

The green and the magenta line were performed with the same SSA as the original simulation but larger asymmetry factors g. Resulting correlation coefficients are clearly smaller.

To conclude, the variation of $O_4$ columns along almucantars contain information about the asymmetry factor g. Interestingly,

the value of g = 0.75 measured by the close-by AERONET station leads to the largest correlation coefficients in Fig. 14d.





However, simulations using smaller asymmetry factors (not plotted) show a similar performance unless g reaches very small values (g < 0.5). Consequently, the simple approach of using correlation coefficients as performed here is not a sufficient way to determine g with good precision. However, the potential of using $O_4$ (ideally together with intensity) in more sophisticated retrievals appears to be promising.

For the two initial research questions it can be concluded:

1. In general, different almucantars recorded simultaneously by IMPACT have slightly different scattering angles meaning that the information content they provide is not redundant. Consequently, these almucantars have a potential to be used in future retrievals of the aerosol phase function. In particular, use of almucantar $O_4$ columns turned out to contain information about the asymmetry factor g, but to be insensitive to the SSA.

2. As a compromise, $10°$ elevation appears to be a reasonable threshold for deriving aerosol phase function information from almucantar $O_4$ measurements. Note, this threshold corresponds to the special conditions during the analysed case study (AOD, aerosol profile, weather and viewing conditions, etc.) as well as the true spatial homogeneity around the measurement location. However, results may be representative for semi-rural sites like Cabauw where the aerosol profile is assumed to be rather spatially constant. Within cities, the spatial variability of aerosols will be much larger and therefore more of the lower almucantars would have to be excluded. As a recipe for unclear aerosol conditions, checking the agreement between measured $O_4$ columns obtained clockwise and counter-clockwise from the SAA (as in Fig. 14b) gives a first indication whether data from the respective elevation angle can be used or not.

## 4.3   NO$_2$ profiling

As already mentioned, one of IMPACT's objectives is to enable aerosol and trace gas profile retrievals very rapidly in every
direction around the measurement site. The retrieval code BOREAS (Bösch et al., 2018) used here is an IUP-Bremen in-house algorithm. Results of the profile inversion help in particular to analyze and support findings of the case studies reported in previous sections.

    For the current study, profiles are retrieved on an altitude grid reaching from 0 to 4 km in 100 m steps. $NO_2$ slant columns in prescribed elevation angles were used for the profile retrieval on MAX-DOAS measurements. For IMPACT, all elevations
from $0.6°$ to $10°$ and $29°$ to $31°$ have been used (while other simultaneously measured elevations have been excluded in order to decrease computational time). As additional input for the retrieval, vertical profiles of pressure and temperature were created by taking the mean of 16 different sonde measurements taken during the years 2013-2015 in De Bilt, the Netherlands. The retrieval is based on an Optimal Estimation Method (OEM), for which an exponentially decreasing apriori profile having a surface concentration of $9.13 \cdot 10^{10}$ molec/cm$^3$ and a scale height of 1 km has been used. For the aerosol profile retrieval,
a surface extinction of 0.183 km$^{-1}$ and again a scaling height of 1 km has been assumed. For the aerosol phase function and SSA, always the closest-in-time values obtained from the near-by Cabauw AERONET station were applied. Radiative transfer calculations were performed using SCIATRAN (Rozanov et al., 2014) in its version 4.0.1. The BOREAS inversion algorithm is explained in detail in Bösch et al. (2018).





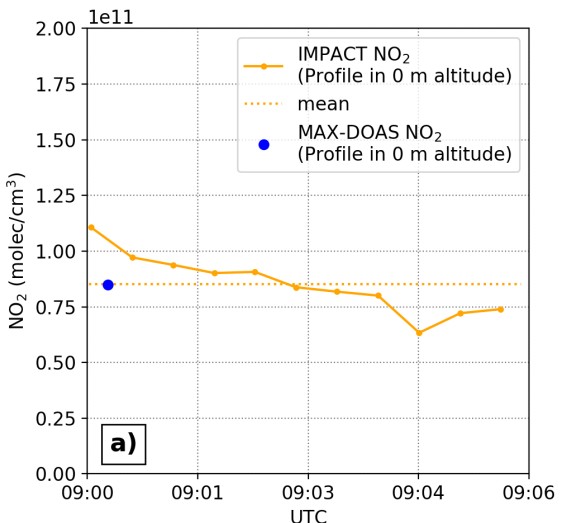
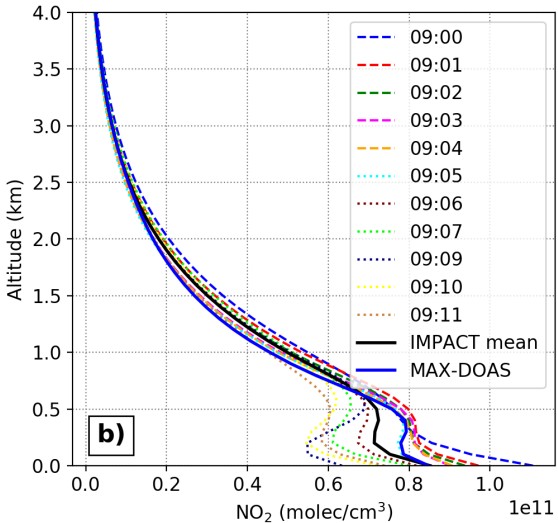

**Figure 15.** $NO_2$ surface concentrations (a) and profiles (b) retrieved from IMPACT's high repetition measurements in the common azimuth direction of 287° during the acquisition of one MAX-DOAS vertical scan. IMPACT and MAX-DOAS $NO_2$ DSCDs used for the BOREAS retrieval are shown in Fig. 6.

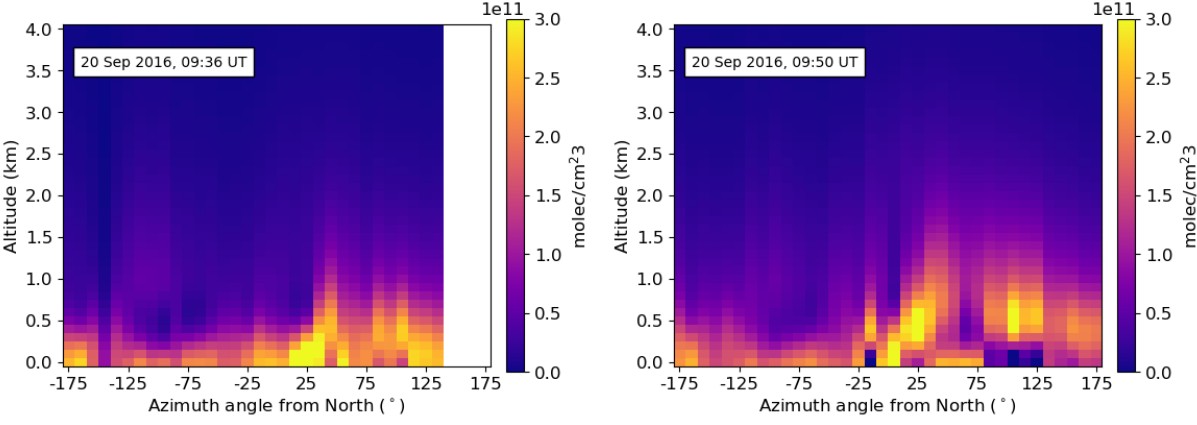

**Figure 16.** Retrieved $NO_2$ profiles around the measurement site during the observed transport event on 20 September 2016.

### 4.3.1 Temporal resolution

As demonstrated in Sect. 3.2, $NO_2$ slant columns can change during the acquisition time of a MAX-DOAS vertical scanning sequence in a fixed azimuth direction (∼20% were observed even under good weather and viewing conditions). If this MAX-DOAS scan is input to a profile retrieval, the change of $NO_2$ is 1) not resolved and 2) possibly interfering the results, predominantly as the retrieved profiles will not simply be a temporal average of the true profiles.



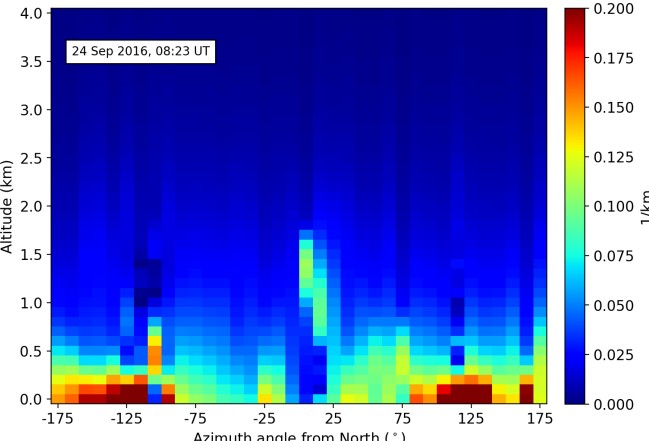

**Figure 17.** Retrieved aerosol extinction profiles around the measurement site for the azimuthal scan shown in Fig. 13.

This is demonstrated in Fig. 15 showing IMPACT and MAX-DOAS surface concentrations and profiles for the case study presented above in Fig. 6. The temporal evolution of $NO_2$ slant columns seen in Fig. 6 is reproduced by $NO_2$ surface concentrations from IMPACT. Interestingly, the change in surface concentrations is even more pronounced and in the order of ≈40% because aerosol concentrations were changing as well. In comparison, the $NO_2$ surface concentration derived from the

single MAX-DOAS profile is of course not reflecting the $NO_2$ decrease, but is (in this case) close to the temporal mean. This is also shown in Fig. 15b comparing single profiles from IMPACT and their mean (solid black line) to the MAX-DOAS profile. However, apart from the surface concentrations, the MAX-DOAS profile and the mean of IMPACT profiles do not agree. Especially in lower altitudes the MAX-DOAS profile is closer to the IMPACT profiles acquired first (between 09:00 and 09:05 UTC). This is reasonable because the MAX-DOAS vertical scanning sequence starts with small elevations, which agree with

lowest elevations of the first (simultaneous) IMPACT scans (see Fig. 6). These small elevations contain much information and have a large influence on the retrieved profile in lower altitudes. In higher altitudes, the information content is limited and the retrieved profile is predominantly determined by apriori information (as discussed in Bösch et al., 2018).

### 4.3.2    $NO_2$ transport event

Full-panoramic $NO_2$ profiles retrieved on 20 September 2016 during the observed transport event (Sect. 4.1) are plotted in

Fig. 16 as a function of azimuth and elevation angle. Viewing conditions during that time were challenging (broken clouds, unstable cloud conditions), affecting the retrieval results. Nevertheless, in agreement with findings in Sect. 4.1, increased $NO_2$ concentrations are observed between azimuths of 25° and -175° from North. As Fig. 16 (left) shows, these increasing concentrations are located close to the ground. The $NO_2$ is then uplifted around 10 UTC (Fig. 16 right) to altitudes of 500-1000 m and in subsequent scans transported in Westerly directions (profiles not shown due to poor viewing conditions). In

general, this is in agreement with findings above and in particular the appearance of high $NO_2$ concentrations close to the



ground and subsequent uplifting supports the conclusion derived in Sect. 4.1 of a local emission source in the vicinity of the measurement site (Lopik or the near-by industrial park).

### 4.3.3  Aerosol inhomogeneity

In Sect. 4.2, enhanced $O_4$ slant columns were found in small elevation angles between -50° and 25° azimuth on 24 September 2016 and it was concluded that these values are caused by smaller aerosol loads close to the ground. Indeed, aerosol extinction profiles retrieved by BOREAS (Fig. 17), show smaller values close to the ground between -50° and 25° azimuth, which supports the conclusions above. However, the BOREAS aerosol retrieval for this day is challenging due to the relatively small absolute aerosol load (AOD $\approx$ 0.2) and consequently Fig. 17 should not be over-interpreted. The general pattern appear reliable, but individual values should be regarded with care.

## 5  Summary and conclusions

An advanced imaging-DOAS instrument (IMPACT) was developed. In contrast to most imaging-DOAS instruments reported so far, IMPACT is not restricted to particular scenes but provides full-azimuthal coverage around the measurement site. Azimuthal pointing is performed stepwise by a motor while observations in 50 elevation angles are performed simultaneously due to the imaging capabilities. As a result, a complete panoramic scan is achieved in ∼15 minutes allowing to retrieve tropospheric trace gas profiles around the measurement site at high temporal resolution. In terms of robustness and flexible setup, IMPACT has similar advantages as state-of-the-art MAX-DOAS instruments as a result of separating indoor (spectrometer) and outdoor (light collecting) parts.

The instrument took part in the CINDI-2 intercomparison field campaign in Cabauw, the Netherlands, in September 2016, where an overall excellent agreement with MAX-DOAS measurements was obtained (correlation > 99% for coinciding observations). In contrast to MAX-DOAS, IMPACT is able to resolve the temporal variation of $NO_2$ slant columns in a fixed azimuth direction, which was observed to be as large as 20% during the time of a MAX-DOAS scan (10-15 minutes) in a case study under good weather and viewing conditions. This temporal variation of $NO_2$ is present in profiles retrieved from IMPACT measurements as well and corresponding surface concentrations of $NO_2$ showed even larger changes of up to 40%. This variation is missed by the MAX-DOAS profile that agrees better with IMPACT profiles acquired first, as a consequence of the scanning sequence which starts with small elevations containing most information.

The azimuthal distribution of $NO_2$ around the measurement site was found to be very homogeneous on a long term scale (campaign average), but highly variable on shorter timescales (snapshots). In small elevations, relative differences of $NO_2$ slant columns up to ∼120% (on average 35%) were observed within one hemispheric scan. In conclusion, measurements in one direction are not enough to characterize tropospheric $NO_2$, which is in particular crucial for MAX-DOAS validation of tropospheric $NO_2$ from satellites.

One reason of the observed $NO_2$ variability are transport events. Due to the fast data acquisition and full azimuthal coverage of IMPACT, the trajectory of an exemplary $NO_2$ transport event could be derived and its most probable source region was



identified in the vicinity of the measurement station (near-by industrial park or village of Lopik). This is supported by BOREAS profile inversions showing increasing $NO_2$ concentrations close to the ground in the azimuthal direction of the trajectory's origin (the assumed source). The $NO_2$ plume is then uplifted and transported along the measurement site in agreement to the trajectory derived before.

The comparison of measured and simulated $O_4$ slant columns demonstrated the huge impact of aerosols on radiative transfer and thus the need to accurately consider them in air mass factor calculations and profile inversions. The azimuthal distribution of $O_4$ columns was found to be sensitive to the asymmetry factor g, and for a test case, a simple trial and error retrieval was performed reproducing the value of g from a near-by AERONET station. As a further advantage, IMPACT is not limited to the solar almucantar as many elevations and therefore several almucantars are measured simultaneously. Each recorded

almucantar observes slightly different scattering angles and provides therefore complementary information. However, care must be taken as for small elevations the influence area (i.e. the spatial region around the measurement site from which information is collected) is increasing. Thus, inhomogeneities of the aerosol distribution around the measurement site were found especially for elevation angles $< 10°$. Consequently, only almucantars $> 10°$ elevation should be used in retrievals of aerosol phase functions. It is important to note that this holds true for specific conditions during CINDI-2 and the spatial

aerosol variability at Cabauw. Nevertheless, Cabauw is believed to be representative for semi-rural environments. For use in different environments, the agreement between $O_4$ columns clockwise and counter-clockwise to the SAA should be checked before corresponding data is used in an aerosol phase function retrieval.

In summary, the added value of full-panoramic imaging-DOAS sensors like IMPACT, in comparison to MAX-DOAS instruments, is predominantly the ability to resolve the spatial and temporal trace gas variability around the measurement site, which

has been demonstrated here for $NO_2$. Thus, as a perspective for future applications, full-panoramic imaging-DOAS sensors have a large potential in particular for satellite validation activities, as for this purpose the variability of trace gases around the measurement site (i.e. within a satellite pixel) is crucial.

*Acknowledgements.* We thank KNMI for organizing and hosting the CINDI-2 campaign and the CESAR test site team for their support and providing helpful complementary data. The MAX-Plank Institute for Chemistry, Mainz, was providing dedicated Xenon lamp measurements

allowing to perform pointing calibration, which was crucial for the analysis of the IMPACT measurements - many thanks in particular to Sebastian Donner, Jonas Kuhn, and Thomas Wagner who operated the lamp for long time periods in the field. We also acknowledge AERONET-Europe/ACTRIS for calibration and maintenance services - the research leading to these results has received funding from European Union's Horizon 2020 research and innovation programme under grant agreement No 654109. For the provision of mean pressure and temperature profiles used within the BOREAS retrieval we thank François Hendrick and Marc Allaart. Financial support was provided

by the University of Bremen and the EU-QA4ECV project. Further financial support through an M8 PostDoc Project from the University of Bremen Institutional Strategy in the framework of the Excellence Initiative is gratefully acknowledged. Mihalis Vrekoussis acknowledges support from the DFG-Research Center/Cluster of Excellence "The Ocean in the Earth System-MARUM". Part of the computations were performed on the HPC cluster Aether at the University of Bremen, financed by DFG in the scope of the Excellence Initiative.



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
