# Peer review of "Full-azimuthal imaging-DOAS observations of $NO_2$ and $O_4$ during CINDI-2"

_Atmospheric Measurement Techniques, 2019_

## Referee Comment (RC1) · Anonymous Referee #1 · 17 May 2019

the comments are in the attached file

Please also note the supplement to this comment:
https://www.atmos-meas-tech-discuss.net/amt-2019-33/amt-2019-33-RC1-supplement.pdf
* * *

---

## Referee Comment (RC2) · Anonymous Referee #2 · 10 Jun 2019

This manuscript introduces a newly developed imaging-DOAS instrument (IMPACT) with the ability to simultaneously measure 50 elevation angles and achieve a panoramic view of the surrounding NO2 distribution within 15 minutes. This enables the retrieval of tropospheric trace gas profiles at high temporal resolution. The observations presented in this paper were made at Cabauw during the CINDI-2 intercomparison campaign and hence, observations made with IMPACT could be compared with coinciding MAX-DOAS measurements. The azimuthal distribution of NO2 around the measurement site was found to be homogeneous on longer time scales but highly variable on short time scales which is certainly of relevance and interest for the validation of tropospheric NO2 from satellites. The authors found that one reason for the observed NO2 variability are transport events and one such event is further investigated in the manuscript. In

addition to the NO2 observations, the potential of O4 measurements along multiple almucantar scans to be used to retrieve information about the aerosol phase function is investigated as well.

The research described in the manuscript is clearly presented and the manuscript is well written. The scientific content is certainly also relevant for AMT and the paper is recommended for publication in AMT.

Specific comments:

Page 2, line 10-11: Sounds a little strange and since the traffic fleet applies to both, domestic and industrial, I would recommend to delete 'in industry,'. And savanna and forest fires can certainly also be anthropogenic (intentional burn-offs), so needs some rewording.

Page 2, line 13: Add comma: 'Overall, the ..'

Page 3, line 7: 'In summary, all previously reported . . .'

Page 3, line 14: '. . . retrieval of the entire . . .'

Page 3, line 15: 'The short acquisition time . . .' – although discussed later, it would be good to add already here how long (15 min).

Page 3, line 26: Better: '. . . be observed by investigating the temporal . . .'

Page 4, line 12: Better: 'The latter part is . . .'

Page 4, line 25-27: Sentence could be a bit improved, e.g.: . . . either measured at a small solar zenith angle (SZA), or taken . . .(sequential), as for the zenith viewing geometry the light path . . . is then short'

Page 4, line 28: Add comma after (Io)

Page 6, lines 1-5: If there are 69 fibres of which are only 50 used, wouldn't the others be a source of straylight in the spectrograph? If so, how is this dealt with?

Page 6, lines 18-19: Add commas after 'instrument' and after '(Sect. 2.4)'

Page 10, lines 8-9: replace 'realize' with either 'note' or 'acknowledge' and add 'the' before 'telescope elevation'

Page 10, line 11: Delete 'promptly'

Page 12, line 21: 'molec cm-2' needs -2 in superscript Caption of Figure 7: Any reason why that particular period (17-23 Sep) was picked and not e.g. the complete campaign period

Page 14, line 13: Replace 'persistence' with 'persisting', right?

Page 14, line 15: Should be 'overall'

Page 14, line 24 etc.: Would be interesting to know how many such transport events could be identified within the campaign period. Could you add that to the discussion?

Page 15, Figure 8 caption: Could you please add here the time period used (i.e. averaged over)? I assume it is the complete campaign period?

Page 17, Figure 12: Would be helpful if the blue arrow head could be bigger; in my printout, it was not really detectable.

Page 20, line 17: Add comma after 'question (1)'

Page 23, line 4: Add 'with' after 'interfering'

---

## Author Comment (AC2) · 21 Jun 2019

**Author's reply to referee #2:**

**The authors thank anonymous referee #2 for his/her efforts in reviewing our manuscript, which clearly improves during the review process. In the following, the reviewer's comments are printed in black, our replies are indicated in blue. Please find both the reviewer's comments and our point-by-point replies below.**

This manuscript introduces a newly developed imaging-DOAS instrument (IMPACT) with the ability to simultaneously measure 50 elevation angles and achieve a panoramic view of the surrounding $NO_2$ distribution within 15 minutes. This enables the retrieval of tropospheric trace gas profiles at high temporal resolution. The observations presented in this paper were made at Cabauw during the CINDI-2 intercomparison campaign and hence, observations made with IMPACT could be compared with coinciding MAXDOAS measurements. The azimuthal distribution of $NO_2$ around the measurement site was found to be homogeneous on longer time scales but highly variable on short time scales which is certainly of relevance and interest for the validation of tropospheric $NO_2$ from satellites. The authors found that one reason for the observed $NO_2$ variability are transport events and one such event is further investigated in the manuscript. In addition to the $NO_2$ observations, the potential of O4 measurements along multiple almucantar scans to be used to retrieve information about the aerosol phase function is investigated as well.

The research described in the manuscript is clearly presented and the manuscript is well written. The scientific content is certainly also relevant for AMT and the paper is recommended for publication in AMT.

Specific comments:

Page 2, line 10-11: Sounds a little strange and since the traffic fleet applies to both, domestic and industrial, I would recommend to delete 'in industry,'. And savanna and forest fires can certainly also be anthropogenic (intentional burn-offs), so needs some rewording.
**The reviewer is correct, we rephrased this sentence to: "Emission sources of $NO_x$ are both, anthropogenic and biogenic, and comprise e.g. the combustion of fossil fuels for domestic heating and cooking, power generation, traffic, as well as savanna and forest fires."**

Page 2, line 13: Add comma: 'Overall, the ..'
**Included.**

Page 3, line 7: 'In summary, all previously reported …'
**Changed.**

Page 3, line 14: '…retrieval of the entire …'
**Changed.**

Page 3, line 15: 'The short acquisition time …' – although discussed later, it would be good to add already here how long (15 min).
**We added this value here.**

Page 3, line 26: Better: '… be observed by investigating the temporal …'
**The reviewer is right, we changed that accordingly.**

Page 4, line 12: Better: 'The latter part is …'
**True, we changed that accordingly.**

Page 4, line 25-27: Sentence could be a bit improved, e.g.: … either measured at a small solar zenith angle (SZA), or taken … (sequential), as for the zenith viewing geometry the light path … is then short'
**We rephrased the text according to the reviewer's suggestion.**

Page 4, line 28: Add comma after (Io)
**Included.**

Page 6, lines 1-5: If there are 69 fibres of which are only 50 used, wouldn't the others be a source of straylight in the spectrograph? If so, how is this dealt with?
**This is a very good point! The reviewer is correct, the unused upper- and lowermost individual fibres, which are not mapped onto the CCD (due to its dimension and the magnification characteristics of the spectrometer), do not increase the used signal, but certainly increase straylight inside the spectrometer and therefore decrease the ratio I_used/I_straylight.**
**In general, straylight is accounted for in the DOAS analysis by means of an intensity offset correction (usually applied in DOAS fits). The straylight corrections was of zeroth order, which was prescribed by CINDI-2 fit settings (see Tab. 1, which is Tab. 2 in the revised manuscript after suggestions from referee #1). However, in future applications, light from unused fibres should be blocked (respective fibres should be blocked at the entrance slit) to reduce potential straylight problems.**
**We addressed this issue by including the intensity offset correction explicitly in the description of the DOAS analysis (Sect. 2.1). In addition, we rephrased the respective paragraph: "However, as a result of the size of the CCD and the magnification characteristics of the spectrometer, light from the upper- and lowermost fibres do not hit the detector (these fibres are imaged outside the detector area), so that only 50 individual fibres are fully mapped on the CCD used here. This is a non-optimal setup as these fibres do not contribute to the used signal, but enhance straylight within the spectrometer. Although straylight effects are compensated by the intensity offset correction in the later DOAS fit (see Sect. 2.1), light from this non-contributing fibres should be blocked in future applications to reduce potential problems with straylight."**

Page 6, lines 18-19: Add commas after 'instrument' and after '(Sect. 2.4)'
**Included, thanks.**

Page 10, lines 8-9: replace 'realize' with either 'note' or 'acknowledge' and add 'the'
before 'telescope elevation'
**The sentence was rephrased accordingly.**

Page 10, line 11: Delete 'promptly'
**Removed.**

Page 12, line 21: 'molec cm-2' needs -2 in superscript Caption of Figure 7: Any reason why that particular period (17-23 Sep) was picked and not e.g. the complete campaign period?
**Thanks, we corrected the superscript. The reason for the limited period in Fig. 7 is data availability. Unfortunately, not the entire intercomparison period could be covered. IMPACT operated from Sep. 16 in the afternoon until Sep. 24 in the morning when a breakdown occurred. Complete days of parallel operation of MAX-DOAS and IMPACT are therefore 17 – 23 September, which is the time period shown here.**

Page 14, line 13: Replace 'persistence' with 'persisting', right?
**Yes, many thanks. We corrected this.**

Page 14, line 15: Should be 'overall'
**No, but the "average of all" (instead of "mean over all") is meant. We corrected this.**

Page 14, line 24 etc.: Would be interesting to know how many such transport events could be identified within the campaign period. Could you add that to the discussion?

This is a good question which is not easy to answer as we did a qualitative analysis of a specific (the largest occurring) transport event, instead of a quantitative retrieval of the number of transport events. Thus, we cannot give a certain number as we did not elaborate a detection algorithm. However, some events like the analyzed one on 20 September are easy to find manually e.g. from Fig. 9 in the manuscript (which is Fig. 1a below) as hints for a transport event are 1) peaks in the (mean) $NO_2$ slant columns and 2) peaks in the azimuthal variations of $NO_2$ meaning that some azimuthal directions are enhanced while others are not. The event on 20 September is by far the strongest observed variation and explains the largest maximum relative differences (Fig. 1b) during the whole campaign. However, on other days like September 19 (Fig. 2a below), 2 other transport events likely occurred around 9:00 UT and 12:00 UT, while for example on September 18 (Fig. 3a below), no transport event at all is seen. However, the 4° elevation is shown here (arbitrarily) and transport events passing the instrument in a closer distance would enhance predominantly measurements at larger elevations and are thus most likely missed. Therefore, a much more comprehensive analysis is required for a quantitative analysis. In addition, enhancements of the light path due to clouds or aerosols would also enhance the $NO_2$ and thus could be miss-interpreted as $NO_2$ transport event.

In conclusion, a detection algorithm for transport events was not elaborated and is difficult to implement. Nevertheless, the reviewer's suggestion is very good as a quantitative analysis retrieving the number of transport events is valuable information for satellite validation activities. We therefore suggest including this activity in the next $NO_2$ intercomparison campaign.

[Figure]

Fig. 1: Original figure from the manuscript (transport event on September 20 in subplot a)

[Figure]

Fig. 2: The same figure, but subplot a) is replaced by September 19.

[Figure]

Fig. 3: The same figure, but subplot a) is replaced by September 18.

Page 15, Figure 8 caption: Could you please add here the time period used (i.e. averaged over)? I assume it is the complete campaign period?

**This is the mean of all available IMPACT panoramic images. Due to instrument problems (as mentioned above) this is unfortunately not the complete semi-blind intercomparison period, but limited to the period from September 16 (afternoon) to September 24 (morning). We added the time period in the revised manuscript.**

Page 17, Figure 12: Would be helpful if the blue arrow head could be bigger; in my printout, it was not really detectable.
**Thanks for this hint, we increased the arrow head.**

Page 20, line 17: Add comma after 'question (1)'
**Included, thanks.**

Page 23, line 4: Add 'with' after 'interfering'
**Added.**

---

## Author Comment (AC1)

**Author's reply to reviewer 1:**

**Review of "Full-azimuthal imaging-DOAS observations of NO $_2$ and O $_4$ during CINDI-2" by E. Peters et al.**

This paper presents a novel imaging-DOAS instrument able to perform panoramic 360° azimuth views. The instrument is presented in details with: 1) a comparison to MAXDOAS instrument during the CINDI-2 campaign (pointing to horizontal/temporal short term NO2 variability), 2) illustration of a rapid plume transport in the rural Cabauw location, and 3) the potential of O4 measurements added value for the aerosols retrieval with the various almucantar geometries measured simultaneously. The scientific content of the paper fits well the scope of AMT and the manuscript is well written and of interest for the community. The large NO2 variabilities seen on short time scales in the remote location is of interest for MAXDOAS and validation studies. I recommend the publication after the suggested revisions.

We thank the reviewer for his efforts and encouraging comments. Please find our point-by-point replies marked in blue below.

Please note, page and line numbers in the discussion paper do not correspond to issues addressed by the reviewer. This is because the review is based on the uploaded file, which was in "2-column AMT" format. The version published in AMTD is in "1-column manuscript"-format, which was a last minute request of Copernicus Publications. Nevertheless, all issues have been addressed.

**General comments:**

Consider moving paragraph 4.2 after 4.3, to present results in a more clear way (as in the introduction and in the conclusions). To improve readability, please add a sentence explaining that different days are selected to present different studies: first the 23/9, to present temporal variations and comparison with MAXDOAS, then 20/9 to illustrate a transport event and finally the 24/9 for exploring the aerosols potential with O4 measurements. It would be nice to also specify wind conditions for each case.

In the revised manuscript, we provide a table summarizing the meteorological conditions during these days. We also clarified in the introduction (last paragraph explaining the structure of the manuscript) that different example days are shown to demonstrate different aspects (giving links to the respective sections). We agree that this will improve readability.

We agree with the reviewer that shifting paragraphs improves readability. We therefore restructured the revised manuscript:

- We moved the comparison to MAX-DOAS data (Sect. 3.1) into Sect. 4 (Results). The comparison is now Sect. 4.1.
- Following the reviewer's suggestion, we exchanged Sect. 4.2 and 4.3 (now Sect. 4.4 and 4.3).
- From formerly Sect. 4.3 we had to include subsection 4.3.1 and Fig. 17 into the text in Sect. 4.2 (now 4.4), because Fig. 17 was a proof of conclusions in the aerosol section and thus cannot be placed before this.

The text in the introduction was adequately adapted to the new structure of the revised manuscript.

Please clarify somewhere the time needed for 1 azimuth image and how "large" it is, i.e. what is the azimuthal "FOV" (compared to the ~40° in elevation) (p. 4, L 2). My understanding is 10 azimuth degrees covered in one azimuth image, and it needs 15 minutes for the 36 steps that covers the 360°? (but P.6, L12: -175 to 175= 350°, so is it 350 or 360°?). 10 steps azimuth (as mentioned in P.6),

but 11 points on figure 6, which covers 12 minutes... please clarify/add a small paragraph on the azimuth "FOV" (512 pixels for 10° azimuths in one image ?) somewhere (as a confrontation to the elevation 4x0.2°=0.8° FOV and total image of 40 to 41° in the vertical). See also detailed question for figure 6.

We apologize, the azimuthal FOV has never been explicitly introduced so that there is a misunderstanding. 10° is not the azimuthal FOV but the step-size in which the telescope is moved. The instantaneous azimuthal FOV is much smaller and comparable to the instantaneous vertical FOV of a single fibre (which is approximately 0.8°). The single fibres are stacked in the vertical and therefore sum up to a complete vertical coverage of approx. 41° while it remains 0.8° in the horizontal dimension. The size of the instantaneous FOV of a single fibre is determined by its dimensions (active area) and the focal length of the objective used.

As this easily leads to confusion and is an important aspect, we clarified this point and rephrased/added paragraphs in the revised manuscript:

1. In Sect. 2.2 we included:

"[...] Light is collected and focused on the light fibre bundle with a commercial objective (1:1.4, focal length 8 mm). The instantaneous FOV of a single fibre is determined by its dimension (active area) and the focal length of the objective and is about 0.8°, both in the horizontal (azimuthal) as well as in the vertical (elevation) direction. As the single fibres are stack up in the vertical dimension, the resulting hypothetical vertical FOV of the entire fibre bundle is ~58°, i.e. all 69 stacked single fibres. The part of the measurements used for the analysis yields a vertical FOV of ~41° (only 50 individual fibres are fully mapped on the CCD). [...]"

- 2. In Sect. 4.2 we clarified bullet point 2 explaining the panoramic scans in detail:
  - "[...] As a result, a full panoramic view (in the azimuth: 36 consecutively performed measurements between -175° to 175° in 10° steps with an azimuthal FOV of  $\approx 0.8°$  for each measurement; in the vertical: 50 simultaneous measurements of  $\approx 0.8°$  vertical FOV each, covering in total  $\approx$ -5° to 36° elevation angle due to the vertical alignment of the single fibres as explained in Sect. 2.2) was recorded every 15 minutes [...]".

Another misunderstanding arises in Fig. 6, where no azimuthal data is shown. This is the comparison to MAX-DOAS measurements, which were performed for 15 min in a fixed azimuth of 287° from North following a prescribed measurement protocol. This is explained in Sect. 2.4 and therefore potentially confusing for the reader. We clarified this in the figure caption of Fig. 6 in the revised manuscript and give explicitly the link to bullet point 1 in Sect 2.4 (CINDI-2).

Please explain in more details what is done for the zenith measurements and for the dark current correction (see specific comments).

We put more explanations in the revised manuscript. A more comprehensive reply is given below at the respective specific comments.

To improve readability of the figures, also consider adding "N", "E", "W" and "S" letters in addition to azimuth angles from the north in figures 8, 10, 13, 16 and 17. We clarified this explicitly ( $N = 0^\circ$ ,  $E = 90^\circ$ ,  $S = 180^\circ$ ,  $W = -90^\circ$ ) at first appearance in figure 8.

**Specific comments and Technical corrections**

- P. 4, fig2: add the x and y label on the figure for improved readability We replaced the example plot and added axes and labels.

- what is done for the zenith measurements? After each azimuth scan, a zenith image that is correcting pixel-by-pixel the azimuth image? Or one zenith after a whole 360° hemispheric measurement? Never mentioned except very slightly in P.4, L4 P. 4, L 15: 41° vertical FOV: in P2 L70 is 40° - check the coherence!

- The zenith reference measurement (for all CCD lines, binned to represent the individual fibres as explained in Sect. 3.1) is performed after each azimuth scan. This means there is a separate reference spectrum for each elevation angle in a panoramic image but it is the same for different azimuth angles within the panoramic image. We explained that in the revised manuscript in Sect. 2.4 (bullet point 2 that describes the measurement strategy for panoramic scans). We also put a note in table 1 (summary of fit settings).
- 2nd addressed issue: As the precise vertical FOV is fractional, we changed all occurrences in the text to "approx. 41°".

- P. 4, L61: "the detector continuous to be illuminated"  $\rightarrow$  continue to be **Thanks, we changed it to "continues".**

- Considering the increased exposure time (P.5, L5 - how much) to decrease impact of the sequential CCD read out, what is done for the dark current correction?

Dark spectra were recorded routinely for every exposure time applied, including the increased exposure times used to compensate the reduction of intensity by the optical filter. It should be mentioned that the increased exposure times (with filter) are not unusually large, but in the range of normal MAX-DOAS exposure times during twilight. Typical values for the exposure time (with filter) are in the order of a few seconds, which is stated in the revised manuscript. The respective dark signal is then subtracted from the measurement (using the correct exposure time), which is now stated in Sect. 2.2 in the revised manuscript. The procedure is mostly the same as for MAX-DOAS measurements.

- P. 5, L46: cite Kreher et al. for the intercomparison period **Cited here.**

P. 6, L26 : remove the acknowledgements in the acknowledgements section.
 We moved the respective acknowledgements into the acknowledgement section (and deleted it here).

- P.6, L31: the 0.2° steps of the telescope are done in elevation, right?  $\rightarrow$  add 0.02° steps vertically to clarify.

Thanks, we clarified this in the revised manuscript.

P. 6, L35 to 41: figure 3b is not very clear in representing these sentences – there are 3 yellow spots in each of the fibers instead of 4
 The reviewer is right, this could be confusing (the plot was just a sketch). We updated figure 3b.

- P. 6, L97: remove point after "Figure. 6 " **Removed.**

- P. 7, L 21: "while IMPACT repeats measurements of the complete elevation angle range". Clarify in which azimuthal direction. Is figure 6 presenting, in 12 minutes, a full 360° IMPACT scan or only scans in the same "main" azimuthal direction than the MAXDOAS? Same question for P 8, L 6 "the closest IMPACT vertical scan (measured simultaneously) was selected" for figure 7. I.e., is it temporal variability in the MAXDOAS viewing direction (what is the wind speed?) or space variability around the MAXDOAS?

No, there was a confusion, please see our explanation above to Fig. 6 (general comments). No azimuthal data is shown here. As mentioned above, we clarified that in the revised manuscript (in Sect. 2.2, Sect. 2.4 as well as in the caption of Fig. 6).

- Table 2/figure 7: why not including the results for 1° elevation, which is the elevation with the steepest decrease in figure 6? Because of explanations in P. 7, L2 to 14? If yes, this will

also have an impact on the profiling comparison of Sect. 4.3, figure 15b... could you quantify/estimate it? Link the statistical results to those from the semi-blind intercomparison.

The difficulty in using 1° elevation for this intercomparison exercise is that 1° is actually not measured by IMPACT due to the imaging approach used in the instrument (approx.  $0.8^{\circ}$  FOV leading to fractional elevation angles). This is mentioned in the text and in the caption of Fig. 6 (and its legend). We therefore interpolated IMPACT results to integer elevations ( $1.0^{\circ}$ ,  $2.0^{\circ}$  etc.), which is also described in the text. These numbers were then compared to MAX-DOAS. In the case of 1° elevation, the closest IMPACT elevation angles are 1.4° (shown in Fig. 6) and ~0.6°, which is influenced by ground effects (especially because of the overlapping of adjacent fibres shown in Fig. 5). The interpolated NO2 slant column is therefore biased and typically smaller than the 1° MAX-DOAS value. To demonstrate this, we included values for 1° elevation to the regression analysis shown in Fig. 7 in the figure below, but left Fig. 7 in the manuscript as it is.

Figure 1: Same as Fig. 7 in the manuscript, but including the (interpolated) 1° elevation slant columns.

Note, for the profile retrieval BOREAS (and radiative transfer calculations therein) the true elevations have been used, i.e. round angles for MAX-DOAS and fractional angles for IMPACT. The results should therefore be free of any bias from interpolating issues. However, depending on the profile shape, differences can occur due to different sampling of the vertical scanning sequence (e.g. MAX-DOAS measures in 1° and 2°, IMPACT in 1.4° and 2.3°, and the profile is retrieved based on these angles). However, differences in elevation angles are usually in the range of 0.5° or less, so that no severe differences are expected for smooth profiles.

The reviewer is correct that we didn't put enough references to the official semi-blind intercomparison paper (Kreher et al., 2019). The reason is that the official intercomparison exercise was not yet published at time of writing, which was the reason to include a separate intercomparison section in our study. However, in general, the results from the official semi-blind intercomparison are similar to our findings, which we mention now in the revised manuscript. In particular, the slope shows the same behavior (close to one for small elevations, largest values of almost up to 1.1 for 15° and a smaller value again for 30°, if compared to Fig. 17 in Kreher et al 2019). Nevertheless, absolute numbers differ, which is expected as we only compare to a single instrument, while in the official intercomparison exercise a reference data set consisting of several instruments is used. In addition, the time periods on which the intercomarison is based, is different in our study and in Kreher et al. 2019.

- P. 8, L 45: "In general, largest NO2 slant columns are found not in 0 or 1 but  $\sim$ 2 elevation,... which is an effect of the instrument's FOV, i.e. surface effects are present in the 0 and (to a

lesser extent) in 1 elevation angle as a result of the overlap of adjacent fibres mapped onto the CCD ": is this taken into account in the profiling? How?

No, surface effects have generally not been taken into account, neither for MAX-DOAS nor IMPACT profile retrievals. However, see also our explanation above and below to p. 14 l.90.

- P. 9, L 10 to 16: it would be nice to compare the horizontal variability during the campaign illustrate in figure 9 for 4° elevation (between 10 and 120%, with 35% in average), to the vertical variability in the first kilometre

We don't understand the point here. The azimuthal variability, which was investigated, is a measure of the homogeneity around the measurement site, which turned out to be more inhomogeneous than expected (which in turn is important for satellite validation activities). The vertical variability on the other hand is largely influenced by convection and local sources/events, and is not easily obtained as profile retrievals for the complete campaign would have to be performed and analyzed. Nevertheless, given the advantages of an imaging instrument, it would be interesting to retrieve the vertical variability in the first km in different azimuths, for example to detect sources. However, this would have to be done in a subsequent analysis.

- P. 9, L 22: cite references of validation studies that did this averaging in several directions. Is averaging ground-based data in time also an advisable option (i.e Pandora instruments measuring with a very high frequency)?

A validation of OMI satellite pixels taking into account not only the azimuthal inhomogeneity around the measurement site but also changes of the NO2 concentration along the light path (using 3D DOAS) was presented by Ortega et al. (2015). However, spatial inhomogeneity (predominantly in the context of satellite validation) is usually regarded in urban areas, where it is expected. The new finding in our study is that even in rural or semi-rural areas like Cabauw, the spatial variability can be much larger than expected, at least partly due to transport events, and has to be considered when performing satellite validation. We point this out more clearly in the revised manuscript.

- Figure 11: add a little bit of description (beta is the azimuth, 75° is the mean wind direction between 10 and 11h, ...)

We included a description of the angles in the figure caption.

- P. 11, L 27: "However, ... " this sentence is strange. Reformulate to something like "with MAXDOAS it is also possible to incorporate O4... as suggested by Wagner..." We agree and changed the sentence accordingly.

- P. 11, L 33: "the aureole region" of the sun ?! Yes. We clarified this in the revised manuscript.

- P. 11, L 52: "For research question (1) it is important that sky radiometers (e.g. within the AERONET network) and current state of the art MAX-DOAS instrument". Modify the "it is important" by "a limitation of"?

We modified according to the reviewer's suggestion.

- P. 11, L 59: replace to "Fig 13, both above and below the ..." **Replaced.**

- P. 11, L79: "short" and "much larger": give an estimation/order of magnitude.

This strongly depends on viewing conditions and aerosol profile, but in first approximation the horizontal extent is scaling with 1/tan(elevation), if only averaging in the boundary layer is considered and the last scattering point is above the boundary layer height, which is added in the revised manuscript. For a more accurate analysis, extensive radiative transfer simulations would

be required. However, it is just a qualitative argument here leading to the retrieval of the respective elevation threshold, which applies to the meteorological conditions for this case study.

P. 11, L 88: change to "this is not the location of largest scattering angles (occurring at ~55° azimuth only)"
 Changed.

- Figure 13: specify somewhere in the text or figure caption that the sun is at 25° elevation and 125° azimuth (fig 13 a))

This is already explicitly mentioned in the text on p.10, l.15 (attention: issue with pages and lines, see remark above): "The position of the sun is clearly visible at  $\sim$ 125° azimuth (Solar Azimuth Angle, SAA) and  $\sim$ 25° elevation."

- P. 13, L 11: remove "again" when specifying the decrease. Before, only 'increase" as been using for describing figure 14 d). Removed.

- P. 13, L 22: remove "interestingly". This is somehow "hoped", no?! that the measurements of the aeronet "g=0.75" value gives the best correlations. We rephrased the text.

Figure 15: panel a) and b) do not cover exactly the same time-period. A) stops before 9h06, while in b) profiles up to 9h11 are presented, and averaged together. Use a more distinct color than black and blue for the IMPACT mean profile and MAXDOAS profile in panel b.
We apologize, there was an issue with the conversion of the decimal time on the x-axis into a more convenient format (the 11 data points from subfigure b are present in subfigure a, but the time was wrong). We corrected this and provide an updated plot in the revised manuscript. We also added seconds to the legend in the right subfigure, to avoid confusion as minutes are too coarse (measurements were performed fast...). Many thanks!

- check that the day is specified in all the figures (not the case in fig 14 and 15).
We now included the day explicitly in the figure captions (figs. 14 and 15 are linked to figs. 13 and 6, where the day was specified).

- P. 14, L. 90: "These small elevations contain much information and have a large influence on the retrieved profile in lower altitudes": cf previous question on the impact of the low elevations of IMPACT being impacted by surface + impact of the different decimal digits of IMPACT elevation instead of round elevations of MAXDOAS?

What are the Degrees of freedom for the profiles coming from the 2 instruments? Are they comparable?

- To the issue of the different decimal digits (as mentioned above): The MAX-DOAS profile retrieval (and radiative transfer simulations therein) were performed with the correct integer MAX-DOAS elevation angles and the profile retrieval on IMPACT data was performed using the correct fractional elevation angles, i.e. no problem caused by interpolating arises. There is a different sampling on the vertical scanning sequence, but usually differences are smaller than 0.5°, which should not lead to different results except for the case of highly structured profiles.
- To the issue of surface impacts: The open question is, how surface effects impact the retrieval, which cannot be answered completely here. For a test run, we omitted all elevations up to 4° in our profile retrieval (as these directions are blocked by trees in some azimuths). In these retrievals, the number of degrees of freedom was clearly smaller, whereas retrieved profiles were surprisingly quite similar. This is remarkable as it implies that for this test case blocked elevation angles have a similar effect on the

retrieved profile as omitting the blocked measurements completely, which seems to be in contrast to the assumption that the lowest elevations are crucial for the retrieval. A possible explanation is that IMPACT's relatively large effective FOV (due to overlapping of single fibres) makes the retrieved profile somewhat insensitive to single measurements, even at low elevations. This should be further investigated in subsequent studies on profiling, but is out of scope (and out of focus) for our current study (and therefore not discussed in the manuscript).

• To the number of degrees of freedom for MAX-DOAS and IMPACT: This number is a bit larger for MAX-DOAS profile results, which is most likely caused by the smaller FOV compared to IMPACT (which has a small FOV of 0.8° for single fibres, but adjacent fibres overlap and therefore increase the "effective FOV"). This is demonstrated in the following plot showing the degrees of freedom in the morning of 24 September 2016.

Figure 2: Degrees of freedom from the NO2 profile retrieval using IMPACT data (blue) and MAX-DOAS data (red) in the "intercomparison exercise azimuth angle" of 287° on 24 September 2016.

- P. 16, L 16 "coinciding observations". word should be attenuated, as the measurements are up to 12 minutes apart.

No, they are not. As can be seen from Fig. 6, there is a complete IMPACT vertical scan (recorded simultaneously) for every MAX-DOAS measurement. Simultaneous IMPACT slant columns were then interpolated to the MAX-DOAS elevation. As a quality criterion, measurements were only compared if they differ by less than 2 minutes, which was explained in the last paragraph of the (former) Sect. 3.1 (now 4.1). Thus, the text was left as is.

- P. 16, L 35 "measurements in one direction are not enough to characterize tropospheric  $NO_2$ , which is in particular crucial for MAXDOAS validation of tropospheric  $NO_2$  from satellites". This is true, but also the low sensitivity of the satellite close to the ground is a "limiting" factor.

Of course, but the sensitivity of satellite measurements is a different topic and out of scope of our study. The finding here is that the spatial variability even in semi-rural environments like Cabauw is much larger than expected and is very likely neglected in satellite validation activities because the NO2 is assumed to be homogeneously distributed.